# Population scale mapping of transposable element diversity reveals links to gene regulation and epigenomic variation

Tim Stuart[1], Steven R Eichten[2], Jonathan Cahn[1], Yuliya V Karpievitch[1], Justin O Borevitz[2], Ryan Lister[1]*

[1]ARC Centre of Excellence in Plant Energy Biology, The University of Western Australia, Perth, Australia; [2]ARC Centre of Excellence in Plant Energy Biology, The Australian National University, Canberra, Australia

**Abstract** Variation in the presence or absence of transposable elements (TEs) is a major source of genetic variation between individuals. Here, we identified 23,095 TE presence/absence variants between 216 Arabidopsis accessions. Most TE variants were rare, and we find these rare variants associated with local extremes of gene expression and DNA methylation levels within the population. Of the common alleles identified, two thirds were not in linkage disequilibrium with nearby SNPs, implicating these variants as a source of novel genetic diversity. Many common TE variants were associated with significantly altered expression of nearby genes, and a major fraction of inter-accession DNA methylation differences were associated with nearby TE insertions. Overall, this demonstrates that TE variants are a rich source of genetic diversity that likely plays an important role in facilitating epigenomic and transcriptional differences between individuals, and indicates a strong genetic basis for epigenetic variation.

*For correspondence: ryan.lister@uwa.edu.au

Competing interests: The authors declare that no competing interests exist.

## Introduction

Transposable elements (TEs) are mobile genetic elements present in nearly all studied organisms, and comprise a large fraction of most eukaryotic genomes. The two types of TEs are retrotransposons, which transpose via an RNA intermediate requiring a reverse transcription reaction, and DNA transposons, which transpose via either a cut-paste or, in the case of Helitrons, a rolling circle mechanism with no RNA intermediate (*Wicker et al., 2007*). TE activity poses mutagenic potential as a TE insertion may disrupt functional regions of the genome. Consequently, safeguard mechanisms have evolved to suppress this activity, including the methylation of cytosine nucleotides (DNA methylation) to produce 5-methylcytosine (mC), a modification that can induce transcriptional silencing of the methylated locus. In *Arabidopsis thaliana* (Arabidopsis), DNA methylation occurs in three DNA sequence contexts: mCG, mCHG, and mCHH, where H is any base but G. Establishment of DNA methylation marks can be carried out by two distinct pathways—the RNA-directed DNA methylation pathway guided by 24 nucleotide (nt) small RNAs (smRNAs), and the DDM1/CMT2 pathway (*Zemach et al., 2013*; *Matzke and Mosher, 2014*). A major function of DNA methylation in Arabidopsis is in the transcriptional silencing of TEs. Mutations in genes essential for DNA methylation establishment or maintenance can lead to a decrease in DNA methylation levels, expression of previously silent TEs, and in some cases transposition (*Mirouze et al., 2009*; *Miura et al., 2001*; *Saze et al., 2003*; *Lippman et al., 2004*; *Jeddeloh et al., 1999*; *Zemach et al., 2013*). In Arabidopsis, TEs are often methylated in all cytosine sequence contexts, in a pattern distinct from DNA methylation in other regions of the genome. Conversely, DNA methylation often occurs in gene bodies exclusively in the CG context and is correlated with gene expression, although this gene-body

methylation appears dispensable (*Bewick et al., 2016*). Many thousands of regions of the Arabidopsis genome have been identified as differentially methylated between different wild Arabidopsis accessions, although the cause and possible function of these differentially methylated regions remains unclear (*Schmitz et al., 2013*).

TEs are thought to play an important role in evolution, not only because of the disruptive potential of their transposition. The release of transcriptional and post-transcriptional silencing of TEs can lead to bursts of TE activity, rapidly generating new genetic diversity (*Vitte et al., 2014*). TEs may carry regulatory information such as promoters and transcription factor binding sites, and their mobilization may lead to the creation or expansion of gene regulatory networks (*Hénaff et al., 2014*; *Bolger et al., 2014*; *Ito et al., 2011*; *Makarevitch et al., 2015*). Furthermore, the transposase enzymes required and encoded by TEs have frequently been domesticated and repurposed as endogenous proteins, such as the *DAYSLEEPER* gene in Arabidopsis, derived from a hAT transposase enzyme (*Bundock and Hooykaas, 2005*). Clearly, the activity of TEs can have widespread and unpredictable effects on the host genome. However, the identification of TE presence/absence variants in genomes has remained difficult to date. It is challenging to identify the structural changes in the genome caused by TE mobilization using current short-read sequencing technologies as these reads are typically mapped to a reference genome, which has the effect of masking structural changes that may be present. However, in terms of the number of base pairs affected, a large fraction of genetic differences between Arabidopsis accessions appears to be due to variation in TE content (*Cao et al., 2011*; *Quadrana et al., 2016*). Therefore, identification of TE variants is essential in order to develop a more comprehensive understanding of the genetic variation that exists between genomes, and of the consequences of TE movement on genome and cellular function.

In order to accurately map the locations of TE presence/absence variants with respect to a reference genome, we have developed a novel algorithm, TEPID (Transposable Element Polymorphism IDentification), which is designed for population studies. We tested our algorithm using both simulated and real Arabidopsis sequencing data, finding that TEPID is able to accurately identify TE presence/absence variants with respect to the Col-0 reference genome. We applied our TE variant identification method to existing genome resequencing data for 216 different Arabidopsis accessions (*Schmitz et al., 2013*), identifying widespread TE variation amongst these accessions and enabling exploration of TE diversity and links to gene regulation and epigenomic variation.

## Results

### Computational identification of TE presence/absence variation

We developed TEPID, an analysis pipeline capable of detecting TE presence/absence variants from paired end DNA sequencing data. TEPID integrates split and discordant read mapping information, read mapping quality, sequencing breakpoints, as well as local variations in sequencing coverage to identify novel TE presence/absence variants with respect to a reference TE annotation (*Figure 1*; see Materials and methods). This typically takes 5–10 min per accession for Arabidopsis genomic DNA sequencing data at 20-40x coverage, excluding the read mapping step. After TE variant discovery has been performed, TEPID then includes a second refinement step designed for population studies. This examines each region of the genome where there was a TE presence identified in any of the analyzed samples, and checks for evidence of this insertion in all other samples. In this way, TEPID leverages TE variant information for a group of related samples to reduce false negative calls within the group. Testing of TEPID using simulated TE variants in the Arabidopsis genome showed that it was able to reliably detect simulated TE variants at sequencing coverage levels commonly used in genomics studies (*Figure 1—figure supplement 1*).

In order to further assess the sensitivity and specificity of TE variant discovery using TEPID, we identified TE variants in the Landsberg *erecta* (L*er*) accession, and compared these with the L*er* genome assembly created using long PacBio sequencing reads (*Chin et al., 2013*). Previously published 100 bp paired-end L*er* genome resequencing reads (*Schneeberger et al., 2011*) were first analyzed using TEPID, enabling identification of 446 TE presence variants (*Figure 1—source data 1*) and 758 TE absence variants (*Figure 1—source data 2*) with respect to the Col-0 reference TE annotation. Reads providing evidence for these variants were then mapped to the L*er* reference genome, generated by de novo assembly using Pacific Biosciences P5-C3 chemistry with a 20 kb insert library

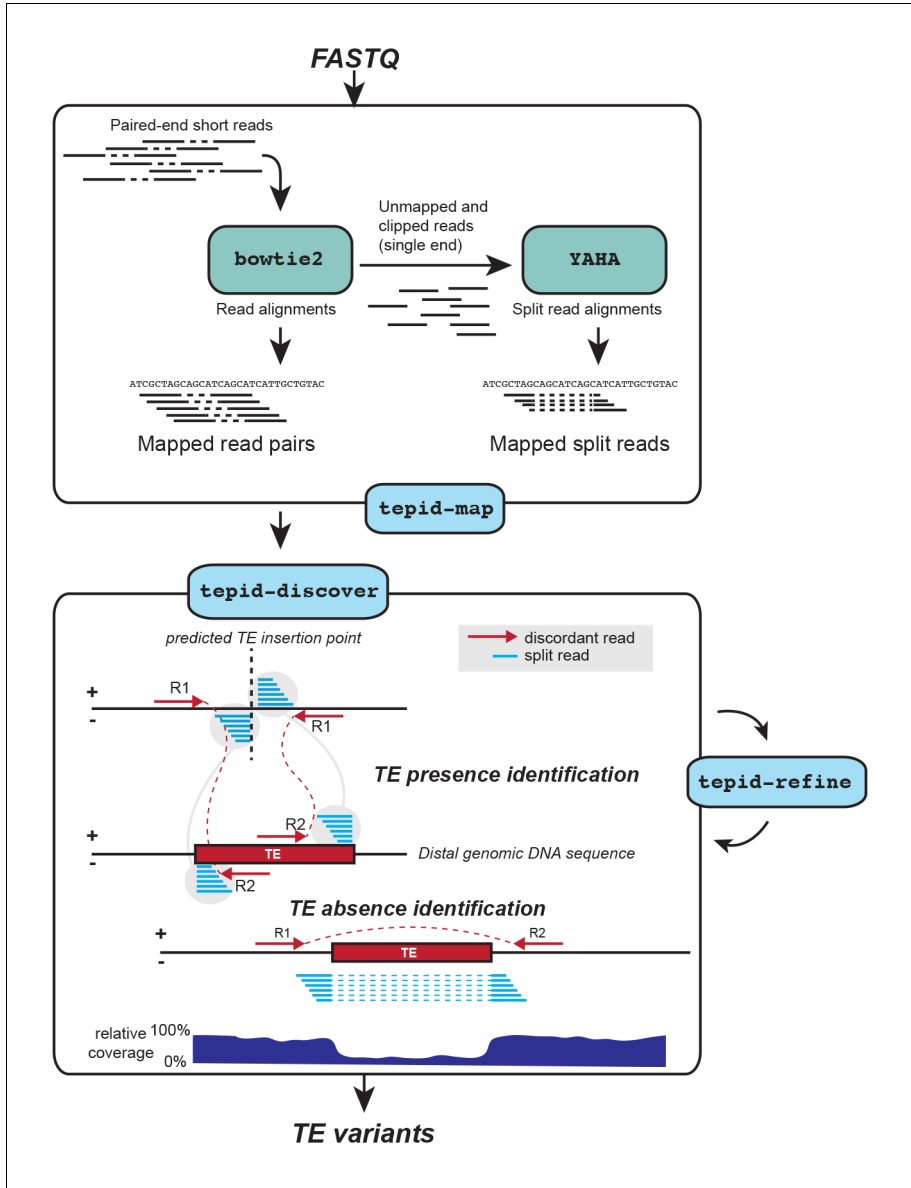

**Figure 1.** TE variant discovery pipeline. Principle of TE variant discovery using split and discordant read mapping positions. Paired end reads are first mapped to the reference genome using Bowtie2 [*Langmead and Salzberg, 2012*]. Soft-clipped or unmapped reads are then extracted from the alignment and re-mapped using Yaha, a split read mapper [*Faust and Hall, 2012*]. All read alignments are then used by TEPID to discover TE variants relative to the reference genome, in the 'tepid-discover' step. When analyzing groups of related samples, these variants can be further refined using the 'tepid-refine' step, which examines in more detail the genomic regions where there was a TE variant identified in another sample, and calls the same variant for the sample in question using lower read count thresholds as compared to the 'tepid- discover' step, in order to reduce false negative variant calls within a group of related samples.

The following source data and figure supplement are available for figure 1:

**Source data 1.** TE presences in L*er.*

**Source data 2.** TE absences in L*er.*

**Figure supplement 1.** Testing of the TEPID pipeline using simulated TE variants in the Arabidopsis Col-0 genome (TAIR10), for a range of sequencing coverage levels.

(*Chin et al., 2013*), using the same alignment parameters as were used to map reads to the Col-0 reference genome. This resulted in 98.7% of reads being aligned concordantly to the L*er* reference, whereas 100% aligned discordantly or as split reads to the Col-0 reference genome (*Table 1*). To find whether reads mapped to homologous regions in both the Col-0 and L*er* reference genomes, we conducted a BLAST search (*Camacho et al., 2009*) using the DNA sequence between read pair mapping locations in the L*er* genome against the Col-0 genome, and found the top BLAST result for 80% of reads providing evidence for TE insertions, and 89% of reads providing evidence for TE absence variants in L*er*, to be located within 200 bp of the TE variant reported by TEPID. Thus, reads providing evidence for TE variants map discordantly or as split reads when mapped to the Col-0 reference genome, but map concordantly to homologous regions of the L*er* de novo assembled reference genome, indicating that structural variation is present at the sites identified by TEPID, and that this is resolved in the de novo assembled genome.

To estimate the rate of false negative TE absence calls made using TEPID, we compared our L*er* TE absence calls to the set of TE absences in L*er* genome identified previously by aligning full-length Col-0 TEs to the L*er* reference using BLAT (*Quadrana et al., 2016*). We found that 89.6% (173/193) of these TE absences were also identified using TEPID, indicating a false negative rate of ~10% for TE absence calls. To determine the rate of false negative TE presence calls, we ran TEPID using 90 bp paired-end Col-0 reads (Col-0 control samples from [*Jiang et al., 2014*]), aligning reads to the L*er* PacBio assembly. As TEPID requires a high-quality TE annotation to discover TE variants, which is not available for the L*er* assembly, we looked for discordant and split read evidence at the known Col-0-specific TEs (*Quadrana et al., 2016*), and found evidence reaching the TEPID threshold for a TE presence call to be made at 89.6% (173/193) of these sites, indicating a false negative rate of ~10%. However, it should be noted that this estimate does not take into account the TEPID refinement step used on large populations, and so the false negative rate for samples analyzed in the population from Schmitz et al. (2013) is likely to be lower than this estimate, as each accession gained on average 4% more presence calls following this refinement step (*Figure 2—figure supplement 1*).

## Abundant TE positional variation among natural Arabidopsis populations

TEPID was used to analyze previously published 100 bp paired-end genome resequencing data for 216 different Arabidopsis accessions (*Schmitz et al., 2013*), and identified 15,007 TE presence variants (*Figure 2—source data 1*) and 8088 TE absence variants (*Figure 2—source data 2*) relative to the Col-0 reference accession, totalling 23,095 unique TE variants. A recent study focused on identifying recent TE insertions containing target site duplications in this population (*Quadrana et al., 2016*). Our goal was to provide a comprehensive assessment of TE presence/absence variation in Arabidopsis. In most accessions TEPID identified 300–500 TE presence variants (mean = 378) and 1000–1500 TE absence variants (mean = 1279), the majority of which were shared by two or more accessions (*Figure 2—figure supplement 2*). Although more TE absences were found on an accession-by-accession basis, overall TE presence variants were more common in the population as the TE absences were often shared between multiple accessions. PCR validations were performed for a random subset of 10 presence and 10 absence variants in 14 accessions (totalling 280 validations), confirming the high accuracy of TE variant discovery using the TEPID package, with a false positive rate for both TE presence and TE absence identification of ~9%, similar to that observed using simulated data and the L*er* genome analysis (*Figure 2—figure supplement 3*). The number of TE presence variants identified was positively correlated with sequencing depth of coverage, while the number of TE

**Table 1.** Mapping of paired-end reads providing evidence for TE presence/absence variants in the L*er* reference genome.

|  | Concordant | Discordant | Split | Unmapped | Total |
|---|---|---|---|---|---|
| **Col-0 mapped** | 0 | 993 | 9513 | 0 | 10,206 |
| **L*er* mapped** | 10,073 | 92 | 34 | 7 | 10,206 |

Note: Discordant and split read categories are not mutually exclusive, as some discordant reads may have one read in the mate pair split-mapped.

absence variants identified had no correlation with sequencing coverage (*Figure 2—figure supplement 4A,B*), indicating that the sensitivity of TE absence calls is not limited by sequencing depth, while TE presence identification benefits from high sequencing depth. However, accessions with low coverage gained more TE presence calls during the TEPID refinement step (*Figure 2—figure supplement 4C*), indicating that these false negatives were effectively reduced by leveraging TE variant information for the whole population.

As TE presence and TE absence calls represent an arbitrary comparison to the Col-0 reference genome, we sought to remove these arbitrary comparisons and classify each variant as a new TE insertion or true deletion of an ancestral TE in the population. To do this, the minor allele frequency (MAF) of each variant in the population was examined, under the expectation that the minor allele is the derived allele. Common TE absences relative to Col-0, absent in ≥80% of the accessions examined, were re-classified as TE insertions in Col-0, and common TE presences relative to Col-0, present in ≥80% of accessions, as true TE deletions in Col-0. Cases where the TE variant had a high MAF (>20%) were unable to be classified, as it could not be determined if these were cases where the variant was most likely to be a true TE deletion or a new TE insertion. While these classifications are not definitive, as there may be rare cases where a true TE deletion has spread through the population and becomes the common allele, it should correctly classify most TE variants. Overall, 72.3% of the TE absence variants identified with respect to the Col-0 reference genome were likely due to a true TE deletion in these accessions, while 4.8% were due to insertions in Col-0 not shared by most other accessions in the population (*Table 2*). High allele frequency TE presence variants relative to Col-0, representing true deletions in Col-0, were much more rare, with 97.8% of initial TEPID TE presence calls being subsequently classified as true insertions. The rarity of true deletions identified in Col-0 is likely due to a reference bias in the TE variant identification method using short read data, as false negative presence calls in the population will reduce the number of true deletions identified in Col-0 due to a reduction in the allele frequency for that variant, causing the frequency of TE presence variants in non-Col-0 accessions to fall below the required 80% threshold for some variants. This is not expected to have a large impact on subsequent population-scale analyses, as Col-0 is only one accession out of the 216 analyzed. Accessions were found to contain on average ~240 true deletions and ~300 true insertions (*Figure 2—figure supplement 5*). Overall, we identified 15,077 TE insertions, 5856 true TE deletions, and 2162 TE variants at a high MAF that were unable to be classified as an insertion or deletion (*Figure 2—source data 3*).

While TE deletions were strongly biased towards the pericentromeric regions where TEs are found in high density, TE insertions had a more uniform distribution over the chromosome. This suggests that TE insertion positions are largely random but may be eliminated from chromosome arms through selection, and accumulate in the pericentromeric regions where low recombination rates prevent their removal (*Figure 2A*). TE deletions and common TE variants were found in similar chromosomal regions, as deletion variants represent the rare loss of common variants. Among TE deletions, DNA TEs were slightly less biased towards the centromeres in comparison to the distribution of RNA TEs (*Figure 2—figure supplement 6*). The distribution of rare (<3% minor allele frequency [MAF], <7 accessions; see Materials and methods) TE variants and TE insertions was similar to that observed for regions of the genome previously identified as being differentially methylated in all DNA methylation contexts (mCG, mCHG, mCHH) between the wild accessions (population C-DMRs) (*Schmitz et al., 2013*). In contrast, population CG-DMRs (differentially methylated in the mCG

**Table 2.** Summary of TE variant classifications.

| TEPID call | TE classification | Count |
|---|---|---|
| Presence | NA | 310 |
| | Insertion | 14,689 |
| | Deletion | 8 |
| Absence | NA | 1852 |
| | Insertion | 388 |
| | Deletion | 5848 |

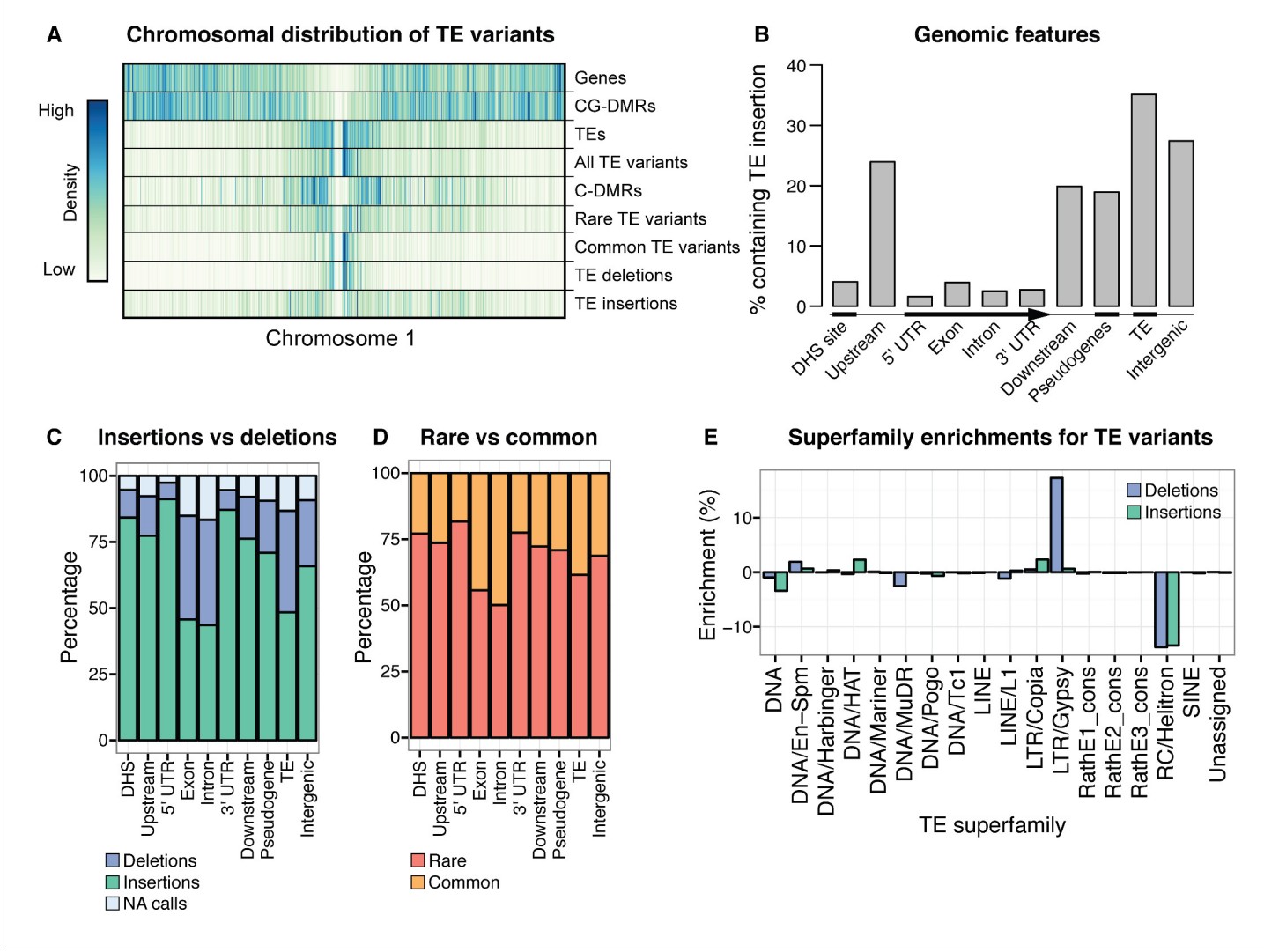

**Figure 2.** Extensive novel genetic diversity uncovered by TE variant analysis. (**A**) Distribution of identified TE variants on chromosome 1, with distributions of all Col-0 genes, Col-0 TEs, and population DMRs. (**B**) Proportion of different genomic features containing one or more TE variants. (**C**) Proportion of TE variants within each genomic feature classified as deletions or insertions. (**D**) Proportion of TE variants within each genomic feature classified as rare (<3% MAF) or common (≥3% MAF). (**E**) Enrichment and depletion of TE variants categorized by TE superfamily compared to the expected frequency due to genomic occurrence.

The following source data and figure supplements are available for figure 2:

**Source data 1.** TE presence variants in all 216 Arabidopsis accessions.

**Source data 2.** TE absence variants in all 216 Arabidopsis accessions.

**Source data 3.** All TE variants.

**Source data 4.** TE family enrichments for TE insertion and TE deletion variants.

**Figure supplement 1.** Percentage of total TE presence calls that were made due to the TEPID refinement step for each accession in the population.

**Figure supplement 2.** Number of accessions sharing TE variants identified by TEPID.

**Figure supplement 3.** Example PCR validations for two TE variants.

*Figure 2 continued on next page*

*Figure 2 continued*

**Figure supplement 4.** Relationship between sequencing depth and number of TE variants discovered in each accession.

**Figure supplement 5.** Number of TE insertions and TE deletions found in each accession.

**Figure supplement 6.** Distribution of RNA and DNA transposable elements over chromosome 1, for TE insertions and TE deletions.

**Figure supplement 7.** Frequency of TE insertion in the *KEE* regions.

**Figure supplement 8.** Length distribution for all Col-0 TEs and all TE variants.

**Figure supplement 9.** TE family enrichments and depletions for TE insertions and TE deletions.

**Figure supplement 10.** TE occupation frequencies for *Basho* TEs previously genotyped by (*Hollister and Gaut, 2007*).

context) less frequently overlapped with all types of TE variants identified and instead closely followed the chromosomal distribution of genes. This was expected, as CG-DMRs are associated with gene bodies whereas C-DMRs are associated with TEs (*Schmitz et al., 2013*). Furthermore, genes and DNase I hypersensitivity sites (putative regulatory regions) (*Sullivan et al., 2014*) rarely contained a TE variant, whereas ~20–35%% of gene flanking regions, pseudogenes, intergenic regions, and other TEs were found to contain a TE variant (*Figure 2B*). This again suggests that TE insertions occur randomly across the genome, with deleterious insertions that occur in functional regions of the genome being subsequently removed through selection. TE deletions and common TE variants were enriched within the set of TE variants found in gene bodies, indicating that TE deletions within genes may be better tolerated than new TE insertions within genes (*Figure 2C,D*). No significant enrichment was found for TE variants within the *KNOT ENGAGED ELEMENT (KEE)* regions, previously identified as regions that may act as a 'TE sink' (*Grob et al., 2014*) (*Figure 2—figure supplement 7*). This may indicate that these regions do not act as a 'TE sink' as has been previously proposed, or that the 'TE sink' activity is restricted to very recent insertions, as the insertions we analysed in this population were likely older than those used in the *KEE* study (*Grob et al., 2014*).

Among the identified TE variants, several TE superfamilies were over- or under-represented compared to the number expected by chance given the overall genomic frequency of different TE types (*Figure 2E*). In particular, both TE insertions and deletions in the RC/Helitron superfamily were less numerous than expected, with an 11.5% depletion of RC/Helitron elements in the set of TE variants. In contrast, TEs belonging to the LTR/Gypsy superfamily were more frequently deleted than expected, with a 17% enrichment in the set of TE deletions. This was unlikely to be due to a differing ability of the detection method to identify TE variants of different lengths, as the TE variants identified had a similar distribution of lengths as all Arabidopsis TEs annotated in the Col-0 reference genome (*Figure 2—figure supplement 8*). These enrichments suggest that the RC/Helitron TEs have been relatively dormant in recent evolutionary history, while the LTR/Gypsy TEs, which are highly enriched in the pericentromeric regions, are frequently lost from the Arabidopsis genome. At the family level, we observed similar patterns of TE variant enrichment or depletion (*Figure 2—figure supplement 9*; *Figure 2—source data 4*). As certain TEs present in Col-0 have previously been genotyped in 47 different accessions, allele frequency data was available for some TEs (*Hollister and Gaut, 2007*), and we compared these previous allele frequency estimates with our estimates based on the short read data. We found a weakly positive linear correlation ($r^2 = 0.3$) between the previous allele frequency estimates for *Basho* family TEs and our allele frequency estimates, which may not be unexpected given the differing population sizes and TE variant detection methods used (*Figure 2—figure supplement 10*).

We further examined Arabidopsis (Col-0) DNA sequencing data from a transgenerational stress experiment to investigate the possible minimum number of generations required for TE variants to arise (*Jiang et al., 2014*). In one of the three replicates subjected to high salinity stress conditions, we identified a single potential TE insertion in a sample following 10 generations of single-seed descent, while no TE variants were identified in any of the three control single-seed descent

replicate sets. However, without experimental validation it remains unclear if this represents a true variant. Therefore, we conclude that TE variants may arise at a rate less than 1 insertion in 60 generations under laboratory conditions. Further experimental work will be required to precisely determine the rate of transposition in Arabidopsis.

## Relationship between TE variants and single nucleotide polymorphisms

Although many thousands of TE variants were identified, they may be linked to the previously identified single nucleotide polymorphisms (SNPs), or unlinked from SNPs across the accessions. This distinction is important, as studies aiming to link epigenetic diversity to genetic variants using only SNPs would fail to detect such a link caused by TE variants if the TE variants are not in LD with SNPs. We tested how frequently common TE variants (>3% MAF; see Materials and methods) were linked to adjacent SNPs to determine when they would represent a previously unassessed source of genetic variation between accessions. SNPs that were previously identified between the accessions (*Schmitz et al., 2013*) were compared to the presence/absence of individual TE variants. For the common TE variants in the population, the nearest flanking 300 SNPs upstream and 300 SNPs downstream of the TE variant site were analyzed for local linkage disequilibrium (LD, $r^2$; see Materials and methods). TE variants were classified as being either 'low', 'mid', or 'high' LD variants by comparing ranked $r^2$ values of TE variant to SNPs against the median ranked $r^2$ value for all between SNP comparisons (SNP-SNP) to account for regional variation in the extent of SNP-SNP LD (*Figure 3A,B*) due to recombination rate variation or selection (*Horton et al., 2012*). The majority (61%) of common TE variants had low LD with nearby SNPs, and represent a source of genetic diversity not previously assessed by SNP-based genotype calling methods (*Figure 3C*). 29% of TE variants displayed high levels of LD and are tagged by nearby SNPs, while only 10% had intermediate levels of LD. We observed a positive correlation between TE variant MAF and LD state, with variants of a high MAF more often classified as high-LD (*Figure 3D*). While the proportion of TE variants classified as high, mid, or low-LD was mostly the same for both TE insertions and TE deletions, TE variants with a high MAF (>20%) that were unable to be classified as either true deletions or as new insertions had a much higher proportion of high-LD variants (*Figure 3E*). This was consistent with the observation that the more common alleles were more often in a high-LD state. TE variants displayed a similar distribution over chromosome 1 regardless of linkage classification (*Figure 3—figure supplement 1*). Overall, this analysis revealed an abundance of previously uncharacterized genetic variation that exists amongst Arabidopsis accessions caused by the presence or absence of TEs, and illustrates the importance of identifying TE variants alongside other genetic diversity such as SNPs.

## TE variants affect gene expression

To determine whether the newly discovered TE variants may affect nearby gene expression, the steady state transcript abundance within mature leaf tissue was compared between accessions with and without TE insertions or deletions, for genes with TE variants located in the 2 kb gene upstream region, 5' UTR, exons, introns, 3' UTR or 2 kb downstream region (*Figure 4A*). While the steady state transcript abundance of most genes appeared to be unaffected by the presence of a TE, 168 genes displayed significant differences in transcript abundance linked with the presence of a TE variant, indicating a role for these variants in the local regulation of gene expression (1% false discovery rate; >2-fold change in transcript abundance; *Figure 4A*, *Figure 4—source data 1*). No functional category enrichments in this set of differentially expressed genes were identified. As rare TE variants may also be associated with a difference in transcript abundance, but were unable to be statistically tested due to their rarity, a burden test for enrichment of rare variants in the extremes of expression was performed (*Zhao et al., 2016*). Briefly, this method counts the frequency of rare variants within each gene expression rank in the population, and aggregates this information over the entire population to determine whether an enrichment of rare variants exists within the gene expression extremes for the population. A strong enrichment for gene expression extremes was observed for TE variants in all gene features tested (*Figure 4B*). While TE variants in gene upstream regions showed a strong enrichment of both high and low gene expression ranks, TE variants in exons or gene downstream regions were more skewed towards low expression ranks than high ranks. Randomization of the accession names removed these enrichments completely (*Figure 4—figure supplement 1*), and there was little difference between TE insertions and TE deletions in the gene

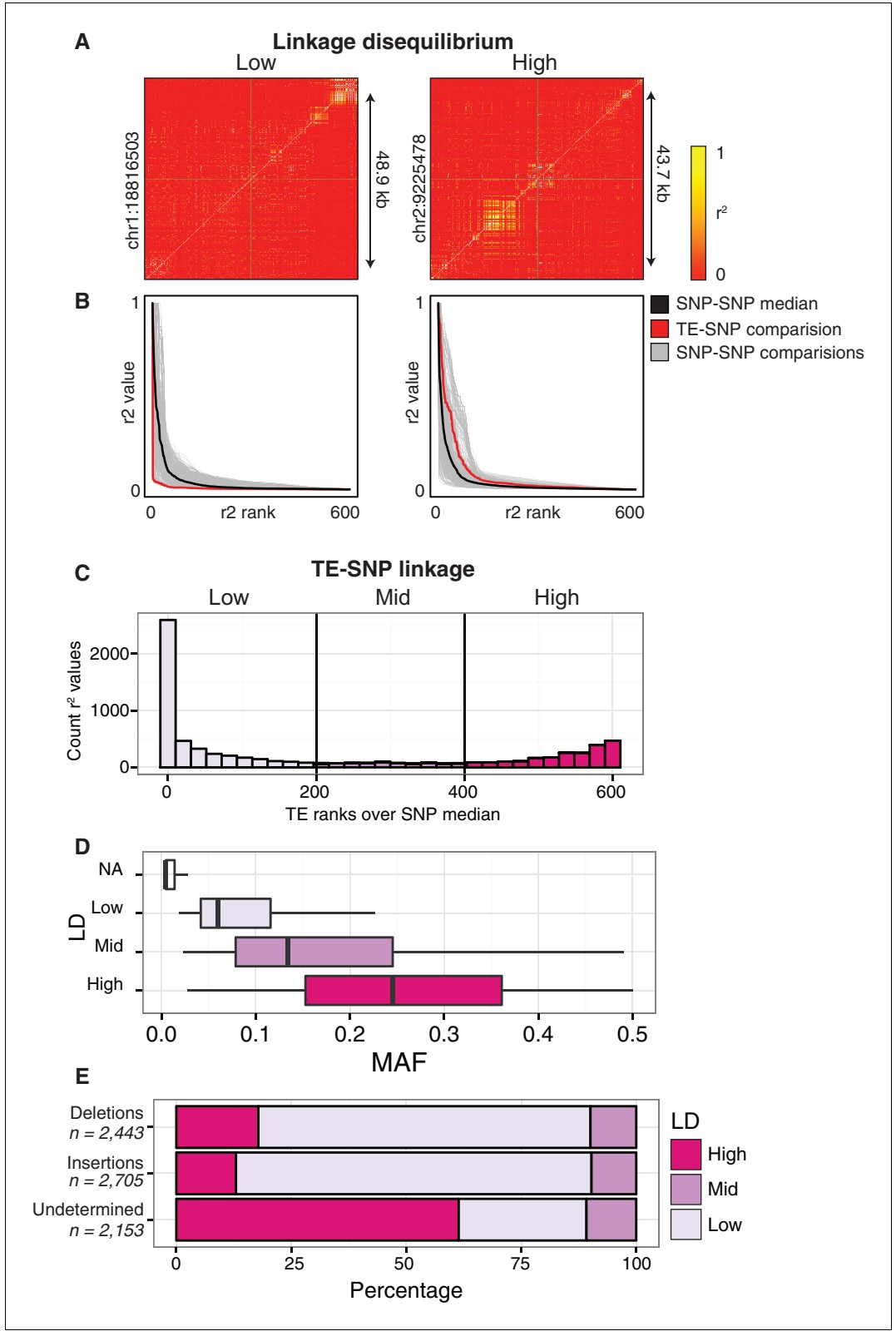

**Figure 3.** Patterns of TE-SNP linkage. (A) $r^2$ correlation matrices for individual representative high and low-LD TE variants showing the background level of SNP-SNP linkage. (B) Rank order plots for individual representative high and low-LD TE variants (matching those shown in A). Red line indicates the median $r^2$ value for each rank across SNP-based values. Blue line indicates $r^2$ values for TE-SNP comparisons. Grey lines indicate all individual SNP-SNP

*Figure 3 continued on next page*

*Figure 3 continued*

comparisons. (C) Histogram of the number of TE $r^2$ ranks (0-600) that are above the SNP-based median $r^2$ value for common TE variants. (D) Boxplots showing distribution of minor allele frequencies for each LD category. Boxes represent the interquartile range (IQR) from quartile 1 to quartile 3. Boxplot upper whiskers represent the maximum value, or the upper value of the quartile 3 plus 1.5 times the IQR (whichever is smaller). Boxplot lower whisker represents the minimum value, or the lower value of the quartile 1 minus 1.5 times the IQR (whichever is larger). (E) Proportion of TE insertions, TE deletions, and unclassified TE variants in each LD category.

The following figure supplement is available for figure 3:

**Figure supplement 1.** Distribution of TE variants across chromosome 1 for each LD category (high, mid, low).

expression rank enrichments found (*Figure 4—figure supplement 2*). This rare variant analysis further indicates that TE variants may alter the transcript abundance of nearby genes, with TE variants in exons or gene downstream regions being mostly associated with gene downregulation, whereas TE variants in gene upstream regions appear to be associated with gene activation and gene repression equally often.

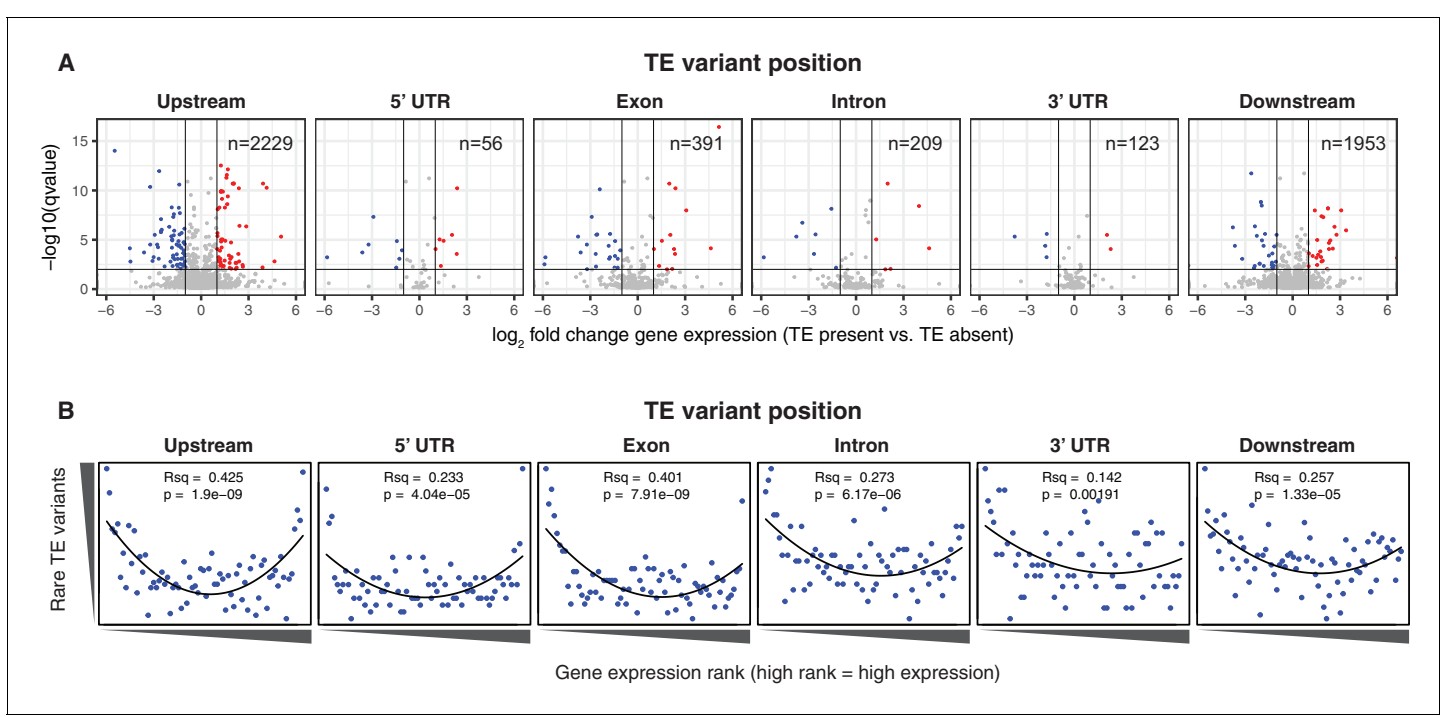

**Figure 4.** Differential transcript abundance associated with TE variant presence/absence. (A) Transcript abundance differences for genes associated with TE insertion variants at different positions, indicated in the plot titles. Genes with significantly different transcript abundance in accessions with a TE insertion compared to accessions without a TE insertion are colored blue (lower transcript abundance in accessions containing TE insertion) or red (higher transcript abundance in accessions containing TE insertion). Vertical lines indicate ±2 fold change in FPKM. Horizontal line indicates the 1% false discovery rate. (B) Relationship between rare TE variant counts and gene expression rank. Cumulative number of rare TE variants in equal-sized bins for gene expression ranks, from the lowest-ranked accession (left) to the highest-ranked accession (right). Lines indicate the fit of a quadratic model.

The following source data and figure supplements are available for figure 4:

**Source data 1.** Differentially expressed genes associated with TE presence/absence.

**Figure supplement 1.** Relationship between rare TE variants and gene expression rank as for *Figure 4B* for permuted TE variants.

**Figure supplement 2.** Relationship between rare TE variants and gene expression rank as for *Figure 4B* for TE insertions and TE deletions separately.

As both increases and decreases in transcript abundance of nearby genes were observed for TE variants within each gene feature, it appears to be difficult to predict the impact that a TE variant may have on nearby gene expression based on TE insertion position alone. Furthermore, gene-level transcript abundance measurements may fail to identify potential positional effects of TE variants upon transcription. To more closely examine changes in transcript abundance associated with TE variants among the accessions, we inspected a subset of TE variant sites and identified TE variants that appear to have an impact on transcriptional patterns beyond simply a change in total transcript abundance of a nearby gene. For example, the presence of a TE insertion within an exon of *AtRLP18* (AT2G15040) was associated with truncation of the transcripts at the TE insertion site in accessions possessing the TE variant, as well as silencing of a downstream gene encoding a leucine-rich repeat protein (AT2G15042) (*Figure 5A,B*). Both genes had significantly lower transcript abundance in accessions containing the TE insertion (p < $5.8 \times 10^{-10}$, Mann-Whitney U test). As four accessions that were predicted to contain the TE insertion within *AtRLP18* appeared to have the non-insertion RNA expression pattern (*Figure 5A*), we performed additional PCR validations on two of these four accessions, as well as two accessions with truncated RNA expression. These validations showed that the accessions predicted to contain the TE insertion but also expressing *AtRLP18* were false positive calls (*Figure 5—figure supplement 1*). However, the false positive rate for this site (~3%) was still lower than our global estimate for TEPID. *AtRLP18* has been reported to be involved in bacterial resistance, with the disruption of this gene by T-DNA insertion mediated mutagenesis resulting in increased susceptibility to the bacterial plant pathogen *Pseudomonas syringae* (*Wang et al., 2008*). Examination of pathogen resistance phenotype data (*Aranzana et al., 2005*) revealed that accessions containing the TE insertion in the *AtRLP18* exon were more often sensitive to infection by *Pseudomonas syringae* transformed with *avrPpH3* genes (*Figure 5C*). This suggests that the accessions containing this TE insertion within *AtRLP18* may have an increased susceptibility to certain bacterial pathogens.

Some TE variants were also associated with increased expression of nearby genes. For example, the presence of a TE within the upstream region of a gene encoding a pentatricopeptide repeat (PPR) protein (AT2G01360) was associated with higher transcript abundance of this gene (*Figure 5D,E*). Transcription appeared to begin at the TE insertion point, rather than the transcriptional start site of the gene (*Figure 5D*). Accessions containing the TE insertion had significantly higher AT2G01360 transcript abundance than the accessions without the TE insertion (p < $1.8 \times 10^{-7}$, Mann-Whitney U test). The apparent transcriptional activation, linked with the presence of a TE belonging to the *HELITRON1* family, indicates that this element may carry regulatory information that alters the expression of genes downstream of the TE insertion site. Importantly, this variant was classified as a low-LD TE insertion, as it is not in LD with surrounding SNPs, and therefore the associated changes in gene transcript abundance would not be linked to genetic differences between the accessions using only SNP data. This TE variant was also upstream of *QPT* (AT2G01350), involved in NAD biosynthesis (*Katoh et al., 2006*), which did not show alterations in transcript abundance associated with the presence of the TE insertion, indicating a potential directionality of regulatory elements carried by the TE (*Figure 5D,E*). This TE insertion occurred at the border of a non-syntenic block of genes thought to be a result of a transposition event in Arabidopsis (*Freeling et al., 2008*). This transposition event likely predates the TE insertion discovered here, and it is interesting that multiple transposition events appear to have occurred in close proximity in the genome. Overall, these examples demonstrate that TE variants can have unpredictable, yet important, effects on the expression of nearby genes, and these effects may be missed by studies focused on genetic variation at the level of SNPs.

## TE variants explain many DNA methylation differences between accessions

As TEs are frequently highly methylated in Arabidopsis (*Zhang et al., 2006*; *Zilberman et al., 2007*; *Cokus et al., 2008*; *Lister et al., 2008*), the DNA methylation state surrounding TE variant sites was assessed to determine whether TE variants might be responsible for differences in DNA methylation patterns previously observed between the wild accessions (*Schmitz et al., 2013*). TE variants were often physically close to DMRs (*Figure 6A*). Furthermore, C-DMRs were more often close to a TE variant than expected, whereas CG-DMRs were rarely close to TE insertions or TE deletions (*Table 3*). Again, this was expected as DNA methylation solely in the CG context is associated with gene

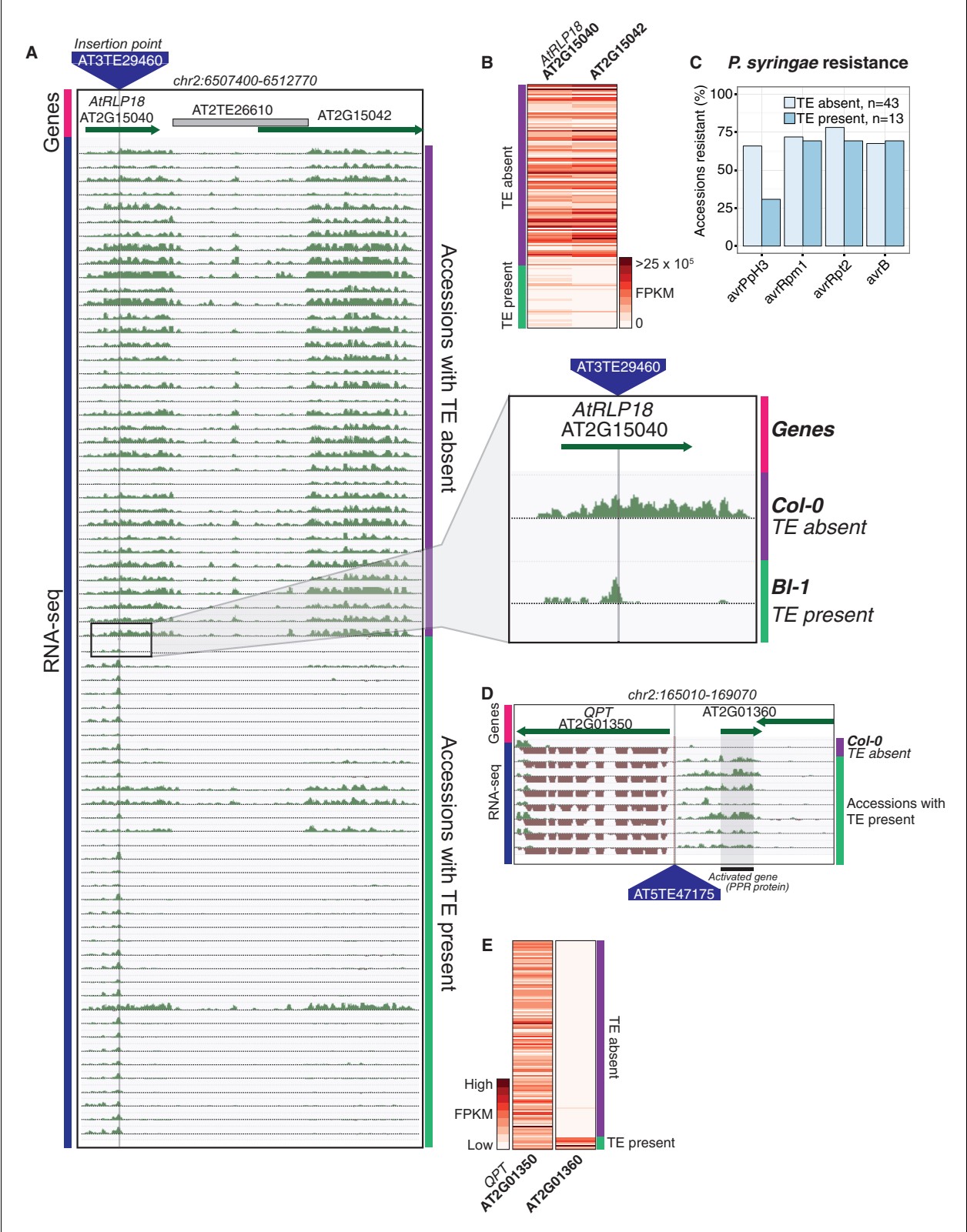

**Figure 5.** Effects of TE variants on local gene expression. (**A**) Genome browser representation of RNA-seq data for genes *AtRLP18* (AT2G15040) and a leucine-rich repeat family protein (AT2G15042). All accessions predicted to contain the TE insertion are shown. Inset shows magnified view of the TE insertion site for two accessions. (**B**) *AtRLP18* and AT2G15042 RNA-seq FPKM values for all accessions. (**C**) Percentage of accessions with resistance to *Pseudomonas syringae* transformed with different *avr* genes, for accessions containing or not containing a TE insertion in *AtRLP18*. (**D**) Genome

*Figure 5 continued on next page*

*Figure 5 continued*

browser representation of RNA-seq data for a PPR protein-encoding gene (AT2G01360) and *QPT* (AT2G01350), showing transcript abundance for these genes in accessions containing a TE insertion variant in the upstream region of these genes, as well as in Col-0. (**E**) RNA-seq FPKM values for *QPT* and a gene encoding a PPR protein (AT2G01360), for all accessions. Note that scales are different for the two heatmaps shown in **E**, due to the higher transcript abundance of *QPT* compared to AT2G01360. Scale maximum for AT2G01350 is $3.1 \times 10^5$, and for AT2G01360 is $5.9 \times 10^4$.

The following figure supplement is available for figure 5:

**Figure supplement 1.** PCR validations for a TE insertion within the *AtRLP18* gene.

bodies, whereas DNA methylation in all contexts is associated with TEs. Overall, 54% of the 13,482 previously reported population C-DMRs were located within 1 kb of a TE variant (predominantly TE insertions), while only 15% of CG-DMRs were within 1 kb of a TE variant (*Table 3*). For C-DMRs, this was significantly more than expected by chance, while it was significantly less than expected for CG-DMRs ($p < 1 \times 10^{-4}$, determined by resampling 10,000 times). Of the C-DMRs that were not close to a TE variant, 3701 (27% of all C-DMRs) were within 1 kb of a non-variable TE. Thus, 81% of C-DMRs are within 1 kb of a TE when considering both fixed and variable TEs in the population. Of the remaining 19% of C-DMRs, most were found in genes or intergenic regions.

To determine whether DMR methylation levels were associated with the presence/absence of nearby TE variants, Pearson correlation coefficients were calculated between the DNA methylation level at each C- or CG-DMR and the presence/absence of the nearest TE variant, to produce a numerical estimate of the association between TE presence/absence and DNA methylation level at the nearest DMR. Further analysis showed that for C-DMRs the strength of this association was dependent on the distance from the C-DMR to the TE insertion, whereas this was not true for CG-DMRs or TE deletions (*Figure 6B*, *Figure 6—figure supplement 1*). This suggested a distance-dependent effect of TE insertion on C-DMR methylation. DNA methylation levels at C-DMRs located within 1 kb of a TE insertion (TE-DMRs) were more often positively correlated with the presence of a TE insertion than the DNA methylation levels at C-DMRs further than 1 kb from a TE insertion (non-TE-DMRs). This was evident from the distribution of correlation coefficients for non-TE-DMRs being centred around zero, whereas for TE-DMRs this distribution was skewed to the right (*Figure 6C*, $D = 0.24$). For TE deletions, such a difference was not observed in the distributions of correlation coefficients between TE-DMRs and non-TE-DMRs, nor for CG-DMRs and their nearby TE insertions or deletions (*Figure 6C*, $D = 0.07–0.10$). These results strongly suggest a relationship between the presence of a TE insertion and formation of a nearby C-DMR.

As the above correlations between TE presence/absence and DMR methylation level rely on the TE variants having a sufficiently high MAF, this precludes analysis of the effect of rare variants on DMR methylation levels. To determine the effect that these rare TE variants may have on DMR methylation levels, a burden test for enrichment of DMR methylation extremes at TE-DMRs was performed, similar to the analysis undertaken to test the effect of rare variants on gene expression. A strong enrichment was observed for high C-DMR and CG-DMR methylation level ranks for TE insertions, while TE deletions were associated with both high and low extremes of DNA methylation levels at C-DMRs, and less so at CG-DMRs (*Figure 6D*). This further indicates that the presence of a TE insertion is associated with higher C-DMR methylation levels, while TE deletions appear to have more variable effects on DMR methylation levels. This enrichment was completely absent after repeating the analysis with randomized accession names (*Figure 6—figure supplement 2*). A slight enrichment was also observed for low DMR methylation ranks for TE insertions near CG-DMRs, indicating that the insertion of a TE was sometimes associated with reduced CG methylation in nearby regions (<1 kb from the TE). Closer examination of these TE insertions revealed that some TE insertions were associated with decreased transcript abundance of nearby genes, with a corresponding loss of gene body methylation, offering a potential explanation for the decreased CG methylation observed near some TE insertions (*Figure 6—figure supplement 3*).

To further assess the effects of TE variants upon local DNA methylation patterns, the levels of methylation were examined in regions flanking all TE variants regardless of the presence or absence of a population DMR call. While DNA methylation levels around pericentromeric TE insertions and deletions (<3 Mb from a centromere) seemed to be unaffected by the presence of a TE insertion

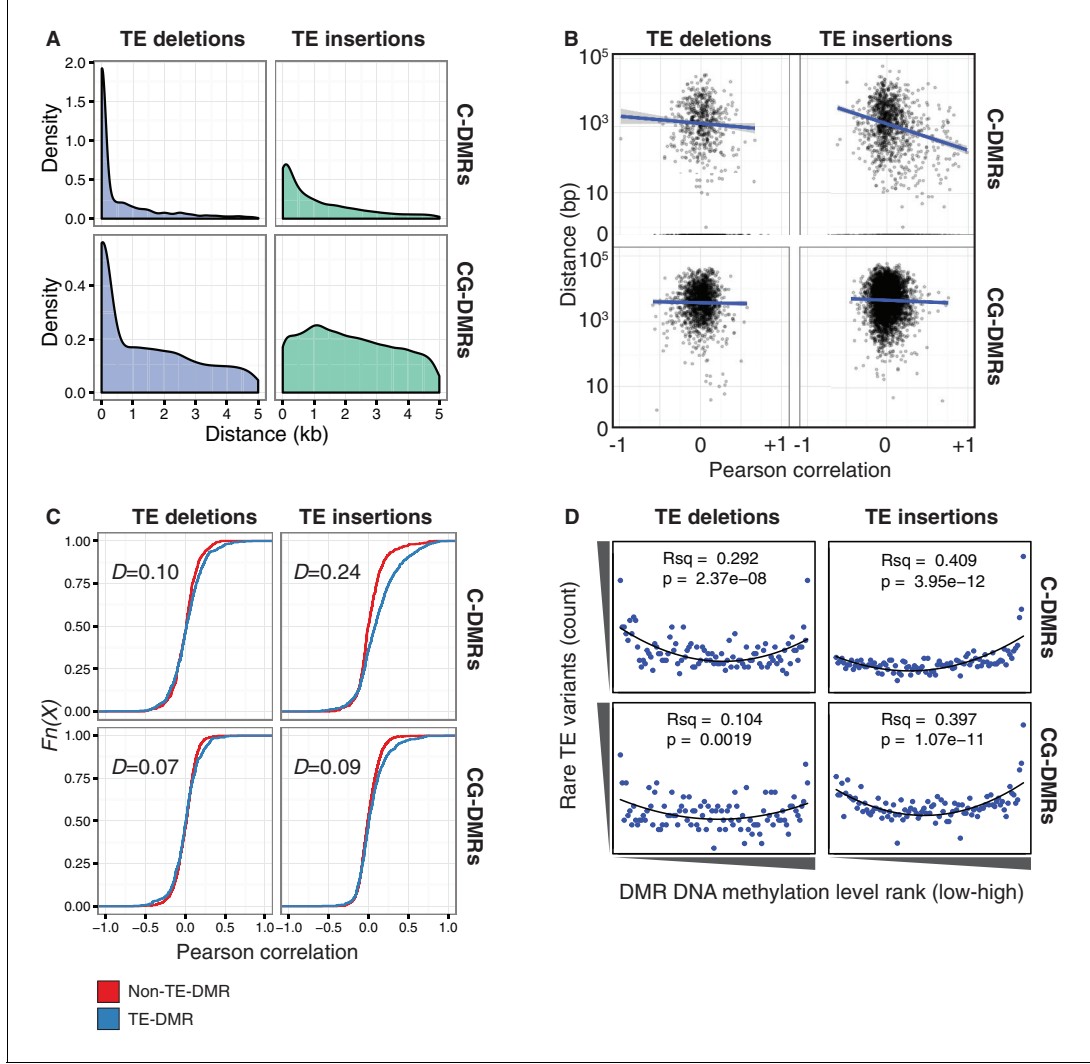

**Figure 6.** TE variants are associated with nearby DMR methylation levels. (**A**) Distribution of distances from TE variants to the nearest population DMR, for TE deletions and TE insertions, C-DMRs and CG-DMRs. (**B**) Pearson correlation between DMR DNA methylation level and TE presence/absence, for all DMRs and their closest TE variant, versus the distance from the DMR to the TE variant (log scale). Blue lines show a linear regression between the correlation coefficients and the log10 distance to the TE variant. (**C**) Empirical cumulative distribution of Pearson correlation coefficients between TE presence/absence and DMR methylation level for TE insertions, TE deletions, C-DMRs and CG-DMRs. The Kolmogorov–Smirnov statistic is shown in each plot, indicated by *D*. (**D**) Relationship between rare TE variant counts and nearby DMR DNA methylation level ranks, for TE insertions, deletions, C-DMRs, and CG-DMRs. Plot shows the cumulative number of rare TE variants in equal-sized bins of DMR methylation level ranks, from the lowest ranked accession (left) to the highest ranked accession (right). Lines indicate the fit of a quadratic model, and the corresponding $R^2$ and p values are shown in each plot.

The following figure supplements are available for figure 6:

**Figure supplement 1.** DNA methylation levels at DMRs near or far from TE variants.

**Figure supplement 2.** Cumulative number DMR methylation level ranks for DMRs near rare TE variants with accessions selected at random.

**Figure supplement 3.** Selected examples of TE insertions apparently associated with transcriptional downregulation of nearby genes and loss of gene body CG methylation leading to the formation of a CG-DMR.

**Table 3.** Percentage of DMRs within 1 kb of a TE variant.

| | C-DMRs | | | CG-DMRs | | |
|---|---|---|---|---|---|---|
| | Observed | Expected | 95% CI | Observed | Expected | 95% CI |
| TE deletions | 17 | 16 | 0.0079 | 4.1 | 16 | 0.0041 |
| TE insertions | 28 | 26 | 0.0089 | 9.1 | 26 | 0.0047 |
| NA calls | 8.7 | 6.2 | 0.0053 | 1.6 | 6.2 | 0.0027 |
| Total | 54 | 48 | 0.01 | 15 | 48 | 0.0054 |

(*Figure 7A*), TE insertions in the chromosome arms were associated with an increase in DNA methylation levels in all sequence contexts (*Figure 7A,B*). In contrast, TE deletions in the chromosome arms did not affect patterns of DNA methylation, as the flanking methylation level in all contexts appeared to remain high following deletion of the TE (*Figure 7A,C*). As the change in DNA methylation levels around most TE variant sites appeared to be restricted to regions <200 bp from the insertion site, DNA methylation levels in 200 bp regions flanking TE variants were correlated with the presence/absence of TE variants. DNA methylation levels were often positively correlated with the presence of a TE insertion when the insertion was distant from a centromere (*Figure 7D*). TE deletions were more variably correlated with local DNA methylation levels, but also showed a bias towards positive correlations for TE deletions distant from the centromeres. However, for TE variants in the chromosome arms the mean correlation between TE insertions and flanking DNA methylation was significantly higher than the mean correlation between TE deletions and flanking DNA methylation (Mann-Whitney U test, p<0.002). As methylome data were available for both leaf and bud tissue for 12 accessions, this analysis was repeated comparing between tissue types, but no differences were observed in the patterns of methylation surrounding TE variant sites between the two tissues (*Figure 7—figure supplement 1*). This suggests that the effect of TE variants upon patterns of DNA methylation may be tissue-independent.

These results indicate that local DNA methylation patterns are influenced by the differential TE content between genomes, and that the DNA methylation-dependent silencing of TEs may frequently lead to the formation of DMRs between wild Arabidopsis accessions. TE insertions appear to be important in defining local patterns of DNA methylation, while DNA methylation levels often remain elevated following a TE deletion, and so are independent from the presence or absence of TEs in these cases. Importantly, the distance from a TE insertion to the centromere appears to have a strong impact on whether an alteration of local DNA methylation patterns will occur. This is likely due to flanking sequences being highly methylated in the pericentromeric regions, and so the insertion of a TE cannot further increase levels of DNA methylation. Overall, a large fraction of the population C-DMRs previously identified between wild accessions are correlated with the presence of local TE variants. CG-DMR methylation levels appear to be mostly independent from the presence/absence of common TE variants, while rare TE variants have an impact on DNA methylation levels at both C-DMRs and CG-DMRs, perhaps due to their more frequent occurrence within the chromosome arms, closer to genes and where CG-DMRs are more abundant (*Figure 2A*).

## Genome-wide association scan highlights distant and local control of DNA methylation

To further investigate the effects of TE variants upon local and distant DNA methylation levels in the genome, an association scan was conducted for all common TE variants (>3% MAF) and all population C-DMRs for the 124 accessions with both DNA methylation and TE variant data available. To test the significance of each pairwise correlation, bootstrap p-value estimates were collected based on 500 permutations of accession labels. TE-DMR associations were deemed significant if they had an association more extreme than all 500 permutations (p<1/500). A band of significant associations was observed for TE insertions and their nearby C-DMRs, signifying a local association between TE insertion presence/absence and C-DMR methylation (*Figure 8A*). This local association was not as strong for TE deletions (*Figure 8B*), consistent with our above findings. While TE variants and DNA methylation showed a local association, it is also possible that TE variation may influence DNA

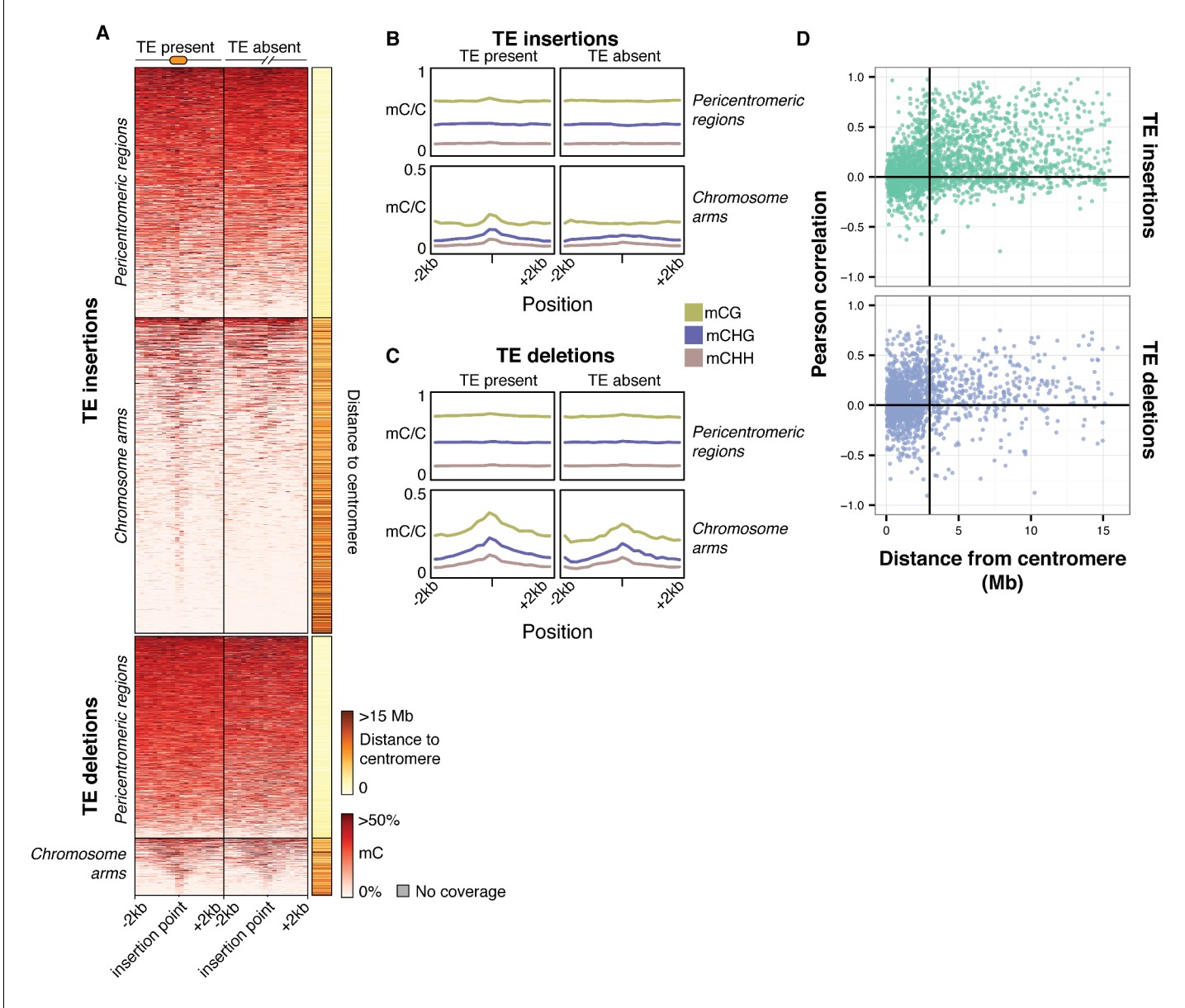

**Figure 7.** Local patterns of DNA methylation surrounding TE variant sites. (**A**) DNA methylation levels in 200 bp bins flanking TE variant sites, ±2 kb from the TE insertion point. TE variants were grouped into pericentromeric variants (<3 Mb from a centromere) or variants in the chromosome arms (>3 Mb from a centromere). (**B**) DNA methylation level in each sequence context for TE insertion sites, ±2 kb from the TE insertion point. (**C**) As for B, for TE deletions. (**D**) Distribution of Pearson correlation coefficients between TE presence/absence and DNA methylation levels in the 200 bp regions flanking TE variant, ordered by distance to the centromere.

The following figure supplement is available for figure 7:

**Figure supplement 1.** DNA methylation levels in 200 bp bins flanking TE variant sites in the 12 accessions with DNA methylation data for both leaf and bud tissue, ±2 kb from the TE insertion point.

methylation states more broadly in the genome, perhaps through the production of *trans*-acting smRNAs or inactivation of genes involved in DNA methylation establishment or maintenance. To identify any potential enrichment of C-DMRs regulated in trans, the total number of significant associations was summed for each TE variant across the whole genome (***Figure 8A and B***, top panels). At many sites, far more significant associations were found than expected due to the false positive

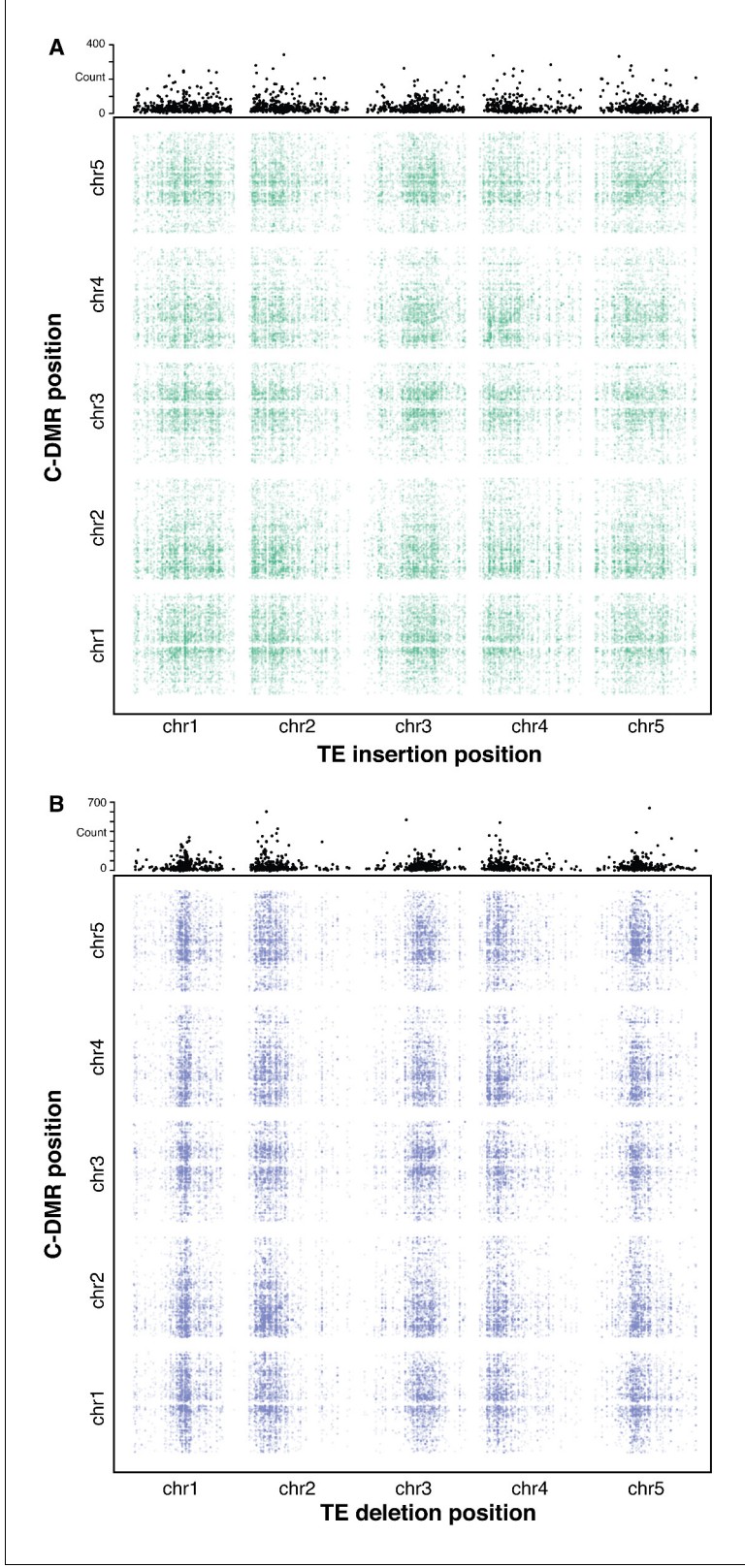

**Figure 8.** Association scan between TE variants and C-DMR methylation variation. (**A**) Significant correlations between TE insertions and C-DMR DNA methylation level. Points show correlations between individual TE-DMR pairs that were more extreme than all 500 permutations of the DMR data. Top plots show the total number of significant correlations for each TE insertion across the whole genome. (**B**) As for (**A**), for TE deletions.

rate alone. This suggested the existence of many putative *trans* associations between TE variants and genome-wide C-DMR methylation levels. These C-DMRs that appeared to be associated with a TE insertion in *trans* were further examined, checking for TE insertions near these C-DMRs that were present in the same accessions as the *trans* associated TE, as these could lead to a false *trans* association. These were extremely rare, with only four such cases for TE insertions, and 38 cases for TE deletions, and so were unable to explain the high degree of *trans* associations found. Overall, this suggests that certain TE variants may affect DNA methylation levels more broadly in the genome, as their effects upon DNA methylation are not necessarily limited to nearby DNA sequences.

## Discussion

Here we have discovered widespread differential TE content between wild Arabidopsis accessions, and explored the impact of these variants upon transcription and DNA methylation at the level of individual accessions. Most TE variants were due to the de novo insertion of TEs, while a smaller subset was likely due to the deletion of ancestral TE copies, mostly around the pericentromeric regions. A subset (32%) of TE variants with a minor allele frequency above 3% were able to be tested for linkage with nearby SNPs. The majority of these TE variants exhibited only low levels of LD with nearby SNPs, indicating that they represent genetic variants currently overlooked in genomic studies. A marked depletion of TE variants within gene bodies and DNase I hypersensitivity sites (putative regulatory regions) is consistent with the more deleterious TE insertions being removed from the population through selection. Of those TE variants found in gene bodies, TE deletions were overrepresented, indicating that the loss of ancestral TEs inserted within genes may be more frequent, or perhaps less deleterious, than the de novo insertion of TEs into genes.

A previous study focused on recent TE insertions in the Arabidopsis population (*Quadrana et al., 2016*), thus the extensive variation between accessions due to older TE insertions or TE deletions has not been explored. We identified clear cases where TE variants appear to have an effect upon gene expression, both in the disruption of transcription and in the spreading or disruption of regulatory information leading to the transcriptional activation of genes, indicating that these TE variants can have important consequences upon the expression of protein coding genes (*Figure 5*). In one case, these changes in gene expression could be linked with phenotypic changes, with accessions containing a TE insertion more frequently sensitive to bacterial infection. Further experiments will be needed to establish a causal link between this TE insertion and the associated phenotype. An analysis of rare TE variants, present at a low MAF, further strengthened this relationship between TE presence/absence and altered transcript abundance, as a strong enrichment of rare TE variants in accessions with extreme gene expression ranks in the population was identified. Therefore, the effects of TE insertions appear to be long-lasting, as there was little difference between common (old) and rare (young) variants in the impact upon gene expression (*Figure 4*).

Perhaps most importantly, we provide evidence that differential TE content between genomes of Arabidopsis accessions underlies a large fraction of the previously reported population C-DMRs. Thus, the frequency of pure epialleles, independent of underlying genetic variation, may be even more rare than previously anticipated (*Richards, 2006*). Overall, 81% of all C-DMRs were within 1 kb of a TE, when considering both fixed and variable TEs in the population. We did not find evidence of CG-DMR methylation, associated with gene bodies, being altered by the presence of common TE variants. However, rare TE variants may be more important in shaping patterns of DNA methylation at some CG-DMRs, perhaps due to their higher frequency in regions close to genes. The level of local DNA methylation changes associated with TE variants was also related to the distance from a TE variant to the centromere, with variants in the chromosome arms being more strongly correlated with DNA methylation levels. This seems to be due to a higher baseline level of DNA methylation at the pericentromeric regions, which prevent any further increase in DNA methylation level following insertion of a TE. Furthermore, we found an important distinction between TE insertions and TE deletions in the effect that these variants have on nearby DNA methylation levels. While flanking DNA methylation levels increase following a TE insertion, the deletion of an ancestral TE was often not associated with a corresponding decrease in flanking DNA methylation levels (*Figure 7*). This indicates that high levels of DNA methylation, once established, may be maintained in the absence of the TE insertion that presumably triggered the original change in DNA methylation level. It is then possible that TE variants explain more of the inter-accession variation in DNA methylation patterns

than we find direct evidence for, if some C-DMRs were formed by the insertion of an ancestral TE that is now absent in all the accessions analysed here. These DMRs would then represent the epigenetic 'scars' of past TE insertions.

Finally, a genome-wide scan of common TE variant association with C-DMR methylation levels provides further evidence of a strong local association between TE insertion presence/absence and C-DMR methylation level (*Figure 8*). The identification of some TE variants that appeared to be associated with changes in DNA methylation levels at multiple loci throughout the genome indicates a possible *trans* regulation of DNA methylation state linked to specific TE variants. Further experiments will be required to confirm and examine the role of these TE variants in determining genome-wide patterns of DNA methylation. Overall, our results show that TE presence/absence variants between wild Arabidopsis accessions not only have important effects on nearby gene expression, but can also have a role in determining local patterns of DNA methylation, and explain many regions of differential DNA methylation previously observed in the population.

## Materials and methods

### TEPID development

#### Mapping

FASTQ files are mapped to the reference genome using the 'tepid-map' algorithm (*Figure 1*). This first calls bowtie2 (*Langmead and Salzberg, 2012*) with the following options: '–local', '–dovetail', '–fr', '-R5', '-N1'. Soft-clipped and unmapped reads are extracted using Samblaster (*Faust and Hall, 2014*), and remapped using the split read mapper Yaha (*Faust and Hall, 2012*), with the following options: '-L 11', '-H 2000', '-M 15', '-osh'. Split reads are extracted from the Yaha alignment using Samblaster (*Faust and Hall, 2014*). Alignments are then converted to bam format, sorted, and indexed using samtools (*Li et al., 2009*).

#### TE variant discovery

The 'tepid-discover' algorithm examines mapped bam files generated by the 'tepid-map' step to identify TE presence/absence variants with respect to the reference genome. Firstly, mean sequencing coverage, mean library insert size, and standard deviation of the library insert size is estimated. Discordant read pairs are then extracted, defined as mate pairs that map more than four standard deviations from the mean insert size from one another, or on separate chromosomes.

To identify TE insertions with respect to the reference genome, split read alignments are first filtered to remove reads where the distance between split mapping loci is less than 5 kb, to remove split reads due to small indels, or split reads with a mapping quality (MAPQ) less than 5. Split and discordant read mapping coordinates are then intersected using pybedtools (*Dale et al., 2011*; *Quinlan and Hall, 2010*) with the Col-0 reference TE annotation, requiring 80% overlap between TE and read mapping coordinates. To determine putative TE insertion sites, regions are then identified that contain independent discordant read pairs aligned in an orientation facing one another at the insertion site, with their mate pairs intersecting with the same TE (*Figure 1*). The total number of split and discordant reads intersecting the insertion site and the TE is then calculated, and a TE insertion predicted where the combined number of reads is greater than a threshold determined by the average sequencing depth over the whole genome (1/10 coverage if coverage is greater than 10, otherwise a minimum of 2 reads). Alternatively, in the absence of discordant reads mapped in orientations facing one another, the required total number of split and discordant reads at the insertion site linked to the inserted TE is set higher, requiring twice as many reads.

To identify TE absence variants with respect to the reference genome, split and discordant reads separated >20 kb from one another are first removed, as 99.9% of Arabidopsis TEs are shorter than 20 kb, and this removes split reads due to larger structural variants not related to TE diversity (*Figure 2—figure supplement 8*). Col-0 reference annotation TEs that are located within the genomic region spanned by the split and discordant reads are then identified. TE absence variants are predicted where at least 80% of the TE sequence is spanned by a split or discordant read, and the sequencing depth within the spanned region is <10% the sequencing depth of the 2 kb flanking sequence, and there are a minimum number of split and discordant reads present, determined by the sequencing depth (1/10 coverage; *Figure 1*). A threshold of 80% TE sequence spanned by split

or discordant reads is used, as opposed to 100%, to account for misannotation of TE sequence boundaries in the Col-0 reference TE annotation, as well as TE fragments left behind by DNA TEs during cut-paste transposition (TE footprints) that may affect the mapping of reads around annotated TE borders (*Plasterk, 1991*). Furthermore, the coverage within the spanned region may be more than 10% that of the flanking sequence, but in such cases twice as many split and discordant reads are required. If multiple TEs are spanned by the split and discordant reads, and the above requirements are met, multiple TEs in the same region can be identified as absent with respect to the reference genome. Absence variants in non-Col-0 accessions are subsequently recategorized as TE insertions present in the Col-0 genome but absent from a given wild accession.

## TE variant refinement

Once TE insertions are identified using the 'tepid-map' and 'tepid-discover' algorithms, these variants can be refined if multiple related samples are analysed. The 'tepid-refine' algorithm is designed to interrogate regions of the genome in which a TE insertion was discovered in other samples but not the sample in question, and check for evidence of that TE insertion in the sample using lower read count thresholds compared to the 'tepid-discover' step. In this way, the refine step leverages TE variant information for a group of related samples to reduce false negative calls within the group. This distinguishes TEPID from other similar methods for TE variant discovery utilizing short sequencing reads. A file containing the coordinates of each insertion, and a list of sample names containing the TE insertion must be provided to the 'tepid-refine' algorithm, which can be generated using the 'merge_insertions.py' script included in the TEPID package. Each sample is examined in regions where there was a TE insertion identified in another sample in the group. If there is a sequencing breakpoint within this region (no continuous read coverage spanning the region), split reads mapped to this region will be extracted from the alignment file and their coordinates intersected with the TE reference annotation. If there are split reads present at the variant site that are linked to the same TE as was identified as an insertion at that location, this TE insertion is recorded in a new file as being present in the sample in question. If there is no sequencing coverage in the queried region for a sample, an 'NA' call is made indicating that it is unknown whether the particular sample contains the TE insertion or not.

While the above description relates specifically to use of TEPID for identification of TE variants in Arabidopsis in this study, this method can be also applied to other species, with the only prerequisite being the annotation of TEs in a reference genome and the availability of paired-end DNA sequencing data.

## TE variant simulation

To test the sensitivity and specificity of TEPID, 100 TE insertions (50 copy-paste transpositions, 50 cut-paste transpositions) and 100 TE absence variants were simulated in the Arabidopsis genome using the RSVSim R package, version 1.7.2 (*Bartenhagen and Dugas, 2013*), and synthetic reads generated from the modified genome at various levels of sequencing coverage using wgsim (*Li et al., 2009*) (https://github.com/lh3/wgsim). These reads were then used to calculate the true positive, false positive, and false negative TE variant discovery rates for TEPID at various sequencing depths, by running 'tepid-map' and 'tepid-discover' using the simulated reads with the default parameters (*Figure 1—figure supplement 1*).

## Estimation of sensitivity

Previously published 100 bp paired end sequencing data for L*er* (http://1001genomes.org/data/ MPI/MPISchneeberger2011/releases/current/Ler-1/Reads/; [*Schneeberger et al., 2011*]) was downloaded and analyzed with the TEPID package to identify TE variants. Reads providing evidence for TE variants were then mapped to the de novo assembled L*er* genome (*Chin et al., 2013*). To determine whether reads mapped to homologous regions of the L*er* and Col-0 reference genome, the de novo assembled L*er* genome sequence between mate pair mapping locations in L*er* were extracted, with repeats masked using RepeatMasker with RepBase-derived libraries and the default parameters (version 4.0.5, http://www.repeatmasker.org). A blastn search was then conducted against the Col-0 genome using the following parameters: '-max-target-seqs 1', '-evalue 1e-6' (*Camacho et al., 2009*). Coordinates of the top BLAST hit for each read location were then compared with the TE variant

sites identified using those reads. To estimate false negative rates for TEPID TE absence calls, Ler TE absence calls were compared with a known set of Col-0-specific TE insertions, absent in Ler (*Quadrana et al., 2016*). For TEPID TE presence calls, we mapped Col-0 DNA sequencing reads (*Jiang et al., 2014*) to the Ler PacBio assembly, and identified sites with read evidence reaching the TEPID threshold for a TE insertion call to be made.

## Arabidopsis TE variant discovery

We ran TEPID, including the insertion refinement step, on previously published sequencing data for 216 different Arabidopsis populations (NCBI SRA SRA012474; [*Schmitz et al., 2013-03*]), mapping to the TAIR10 reference genome and using the TAIR9 TE annotation. The '–mask' option was set to mask the mitochondrial and plastid genomes. We also ran TEPID using previously published transgenerational data for salt stress and control conditions (NCBI SRA SRP045804; [*Jiang et al., 2014*]), again using the '–mask' option to mask mitochondrial and plastid genomes, and the '–strict' option for highly related samples.

## TE variant / SNP comparison

SNP information for 216 Arabidopsis accessions was obtained from the 1001 genomes data center (http://1001genomes.org/data/Salk/releases/2013_24_01/; [*Schmitz et al., 2013-03*]). This was formatted into reference (Col-0 state), alternate, or NA calls for each SNP. Accessions with both TE variant information and SNP data were selected for analysis. Hierarchical clustering of accessions by SNPs as well as TE variants were used to identify essentially clonal accessions, as these would skew the SNP linkage analysis. A single representative from each cluster of similar accessions was kept, leading to a total of 187 accessions for comparison. For all other analyses, the full set of accessions were used in order to maximize sample sizes. For each TE variant with a minor allele frequency greater than 3% (>5 accessions for the SNP linkage analysis), the nearest 300 upstream and 300 downstream SNPs with a minor allele frequency greater than 3% were selected. Pairwise genotype correlations ($r^2$ values) for all complete cases were obtained for SNP-SNP and SNP-TE variant states. $r^2$ values were then ordered by decreasing rank and a median SNP-SNP rank value was calculated. For each of the 600 ranked surrounding positions, the number of times the TE rank was greater than the SNP-SNP median rank was calculated as a relative LD metric of TE to SNP. TE variants with less than 200 ranks over the SNP-SNP median were classified as low-LD insertions. TE variants with ranks between 200 and 400 were classified as mid-LD, while TE variants with greater than 400 ranks above their respective SNP-SNP median value were classified as variants in high LD with flanking SNPs.

## PCR validations

### Selection of accessions to be genotyped

To assess the accuracy of TE variant calls in accessions with a range of sequencing depths of coverage, we grouped accessions into quartiles based on sequencing depth of coverage and randomly selected a total of 14 accessions for PCR validations from these quartiles. DNA was extracted for these accessions using Edward's extraction protocol (*Edwards et al., 1991*), and purified prior to PCR using AMPure beads.

### Selection of TE variants for validation and primer design

Ten TE insertion sites and 10 TE absence sites were randomly selected for validation by PCR amplification. Only insertions and absence variants that were variable in at least two of the fourteen accessions selected to be genotyped were considered. For insertion sites, primers were designed to span the predicted TE insertion site. For TE absence sites, two primer sets were designed; one primer set to span the TE, and another primer set with one primer annealing within the TE sequence predicted to be absent, and the other primer annealing in the flanking sequence (*Figure 2—figure supplement 3*). Primer sequences were designed that did not anneal to regions of the genome containing previously identified SNPs in any of the 216 accessions (*Schmitz et al., 2013-03*) or small insertions and deletions, identified using lumpy-sv with the default settings (*Layer et al., 2014*) (https://github.com/arq5x/lumpy-sv), had an annealing temperature close to 52°C calculated based on nearest neighbor thermodynamics (MeltingTemp submodule in the SeqUtils python module; [*Cock et al., 2009*]), GC content between 40% and 60%, and contained the same base repeated not more than

four times in a row. Primers were aligned to the TAIR10 reference genome using bowtie2 (*Langmead and Salzberg, 2012*) with the '-a' flag set to report all alignments, and those with more than five mapping locations in the genome were then removed.

## PCR

PCR was performed with 10 ng of purified Arabidopsis DNA using Taq polymerase. PCR products were analysed by agarose gel electrophoresis. Col-0 was used as a positive control, water was added to reactions as a negative control.

## mRNA analysis

Processed mRNA data for 144 wild Arabidopsis accessions were downloaded from NCBI GEO GSE43858 (*Schmitz et al., 2013*). To find differential gene expression dependent on TE presence/absence variation, we first removed transposable element genes from the set of TAIR10 gene models, then filtered TE variants to include only those where the TE variant was shared by at least seven accessions with RNA data available. We then grouped accessions based on TE presence/absence variants, and performed a Mann-Whitney U test to determine differences in RNA transcript abundance levels between the groups. We used q-value estimation to correct for multiple testing, using the R qvalue package v2.2.2 with the following parameters: lambda = seq(0, 0.6, 0.05), smooth.df=4 (*Storey and Tibshirani, 2003*). Genes were defined as differentially expressed where there was a greater than two fold difference in expression between the groups, with a q-value less than 0.01. Gene ontology enrichment analysis was performed using PANTHER (http://pantherdb.org).

## DNA methylation data analysis

Processed base-resolution DNA methylation data for wild Arabidopsis accessions were downloaded from NCBI GEO GSE43857 (*Schmitz et al., 2013*), and used to construct MySQL tables in a database.

## Rare variant analysis

To assess the effect of rare TE variants on gene expression or DMR DNA methylation levels, we tested for a burden of rare variants (<3% MAF, <7 accessions) in the population extremes, essentially as described previously (*Zhao et al., 2016*). For each rare TE variant near a gene or DMR, we ranked the gene expression level or DMR DNA methylation level for all accessions in the population, and tallied the ranks of accessions containing a rare variant. These rank counts were then binned to produce a histogram of the distribution of ranks. We then fit a quadratic model to the counts data, and calculated the $R^2$ and p-value for the fit of the model.

## TE variant and DMR genome-wide association analysis

Accessions were subset to those with both leaf DNA methylation data and TEPID calls. Pairwise correlations were performed for observed data pairs for each TE variant and a filtered set of population C-DMRs, with those C-DMRs removed where more than 15% of the accessions had no coverage. This amounted to a final set of 9777 C-DMRs. Accession names were then permuted to produce a randomized dataset, and pairwise correlations again calculated. This was repeated 500 times to produce a distribution of expected Pearson correlation coefficients for each pairwise comparison. Correlation values more extreme than all 500 permutations were deemed significant.

## Data access

TEPID source code can be accessed at http://doi.org/10.5281/zenodo.167274. Code and data needed to reproduce this analysis can be found at https://doi.org/10.5281/zenodo.168094. L*er* TE variants are available in *Figure 1—source data 1* and *2*. TE variants identified among the 216 wild Arabidopsis accessions resequenced by Schmitz et al. (2013) are available in *Figure 2—source datas 1*, *2* and *3*. Source data are available on Dryad (10.5061/dryad.187b3). A genome browser displaying all TE variants can be found at http://plantenergy.uwa.edu.au/~lister/annoj/browser_te_variants.html.

## Acknowledgements

This work was supported by the Australian Research Council (ARC) Centre of Excellence program in Plant Energy Biology CE140100008 (JOB, RL). RL was supported by an ARC Future Fellowship (FT120100862) and Sylvia and Charles Viertel Senior Medical Research Fellowship, and work in the laboratory of RL was funded by the Australian Research Council. TS was supported by the Jean Rogerson Postgraduate Scholarship. SRE was supported by an Australian Research Council Discovery Early Career Research Award (DE150101206). We thank Robert J Schmitz, Mathew G Lewsey, Ronan C O'Malley, and Ian Small for their critical reading of the manuscript, and Kevin Murray for his helpful comments regarding the development of TEPID. We would also like to thank Brandon Gaut for kindly providing Basho TE allele frequency estimates.

## Additional information

### Funding

| Funder | Grant reference number | Author |
| --- | --- | --- |
| Centre of Excellence in Plant Energy Biology, Australian Research Council | CE140100008 | Tim Stuart<br>Steven R Eichten<br>Jonathan Cahn<br>Yuliya Karpievitch<br>Justin Borevitz<br>Ryan Lister |
| Australian Research Council | | Tim Stuart<br>Steven R Eichten<br>Jonathan Cahn<br>Yuliya Karpievitch<br>Justin Borevitz<br>Ryan Lister |
| Sylvia and Charles Viertel Charitable Foundation | | Ryan Lister |
| Australian Research Council | FT120100862 | Ryan Lister |
| Australian Research Council | DE150101206 | Steven R Eichten |

The funders had no role in study design, data collection and interpretation, or the decision to submit the work for publication.

### Author contributions

TS, Conception and design, Acquisition of data, Analysis and interpretation of data, Drafting or revising the article; SRE, Analysis and interpretation of data, Drafting or revising the article; JC, Performed PCR validations of TE variants, Acquisition of data; YVK, Provided statistical guidance, Analysis and interpretation of data; JOB, Conception and design, Analysis and interpretation of data, Drafting or revising the article; RL, Conception and design, Drafting or revising the article, Analysis and interpretation of data

### Author ORCIDs

Tim Stuart, http://orcid.org/0000-0002-3044-0897
Steven R Eichten, http://orcid.org/0000-0003-2268-395X
Jonathan Cahn, http://orcid.org/0000-0002-5006-741X
Ryan Lister, http://orcid.org/0000-0001-6637-7239

## Additional files

### Major datasets

The following dataset was generated:

| Author(s) | Year | Dataset title | Dataset URL | Database, license, and accessibility information |
|---|---|---|---|---|
| Stuart T, Eichten SR, Cahn J, Karpievitch Y, Borevitz JO, Lister R | 2016 | Population scale mapping of transposable element diversity reveals links to gene regulation and epigenomic variation | http://dx.doi.org/10.5061/dryad.187b3 | Available at Dryad Digital Repository under a CC0 Public Domain Dedication |

The following previously published datasets were used:

| Author(s) | Year | Dataset title | Dataset URL | Database, license, and accessibility information |
|---|---|---|---|---|
| Schmitz RJ, Schultz MD, Urich MA , Nery JR, Pelizzola M, Libiger O, Alix A, McCosh RB, Chen H, Schork NJ, Ecker JR | 2013 | Patterns of population epigenomic diversity | http://www.ncbi.nlm.nih.gov/sra?term=SRA012474 | Publicly available at the NCBI Short Read Archive (accession no: SRA012474) |
| Schmitz RJ, Schultz MD, Urich MA, Nery JR, Pelizzola M, Libiger O, Alix A, McCosh RB, Chen H, Schork NJ, Ecker JR | 2013 | Patterns of population epigenomic diversity | http://www.ncbi.nlm.nih.gov/geo/query/acc.cgi?acc=GSE43858 | Publicly available at the NCBI Gene Expression Omnibus (accassion no: GSE43858) |
| Schmitz RJ, Schultz MD, Urich MA, Nery JR, Pelizzola M, Libiger O, Alix A, McCosh RB, Chen H, Schork NJ, Ecker JR | 2013 | Patterns of population epigenomic diversity | http://www.ncbi.nlm.nih.gov/geo/query/acc.cgi?acc=GSE43857 | Publicly available at the NCBI Gene Expression Omnibus (accassion no: GSE43857) |
| Schneeberger K, Ossowski S, Ott F, Klein JD, Wang X, Lanz C, Smith LM, Cao J, Fitz J, Warthmann N, Henz SR, Huson DH, Weigel D | 2011 | Reference-guided assembly of four diverse Arabidopsis thaliana genomes | http://1001genomes.org/data/MPI/MPISchneeberger2011/releases/current/Ler-1/Reads/ | Available at the 1001 Genomes Project |
| Jiang C, Mithani A, Belfield EJ, Mott R, Hurst LD, Harberd NP | 2014 | Environmentally responsive genome-wide accumulation of de novo Arabidopsis thaliana mutations and epimutations | http://www.ncbi.nlm.nih.gov/sra/?term=SRP045804 | Publicly available at the NCBI Short Read Archive (accession no: SRP045804) |
| Sullivan A, Arsovski AA, Lempe J, Bubb KL, Weirauch MT, Sabo PJ, Sandstrom R, Thurman RE, Neph S, Reynolds AP, Stergachis AB, Vernot B, Johnson AK, Haugen E, Sullivan S, Thompson A, Neri FV, Weaver M, Diegel M, Mnaimneh S, Yang A, Hughes T, Nemhauser JL, Queitsch C, Stamatoyannopoulos JA | 2014 | Mapping and Dynamics of Regulatory DNA and Transcription Factor Networks in A. thaliana | https://www.ncbi.nlm.nih.gov/geo/query/acc.cgi?acc=GSM1289359 | Publicly available at the NCBI Gene Expression Omnibus (accession no: GSE53322) |

| Aranzana MJ, Kim S, Zhao K, Bakker E, Horton M, Jakob K, Lister C, Molitor J, Shindo C, Tang C, Toomajian C, Traw B, Zheng H, Zheng H, Bergelson J, Dean C, Marjoram P, Nordborg M | 2005 | Genome-Wide Association Mapping in Arabidopsis Identifies Previously Known Flowering Time and Pathogen Resistance Genes | http://dx.doi.org/10.1371/journal.pgen.0010060.sd002 | Available at PLOS |

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
