## [Decision Letter]

Thank you for submitting your article "Population scale mapping of transposable element diversity reveals links to gene regulation and epigenomic variation" for consideration by *eLife*. Your article has been favorably evaluated by Detlef Weigel (Senior Editor) and two reviewers, one of whom is a member of our Board of Reviewing Editors. The following individual involved in review of your submission has agreed to reveal his identity: Jeffrey Ross-Ibarra (Reviewer #2).

The reviewers have discussed the reviews with one another and the Reviewing Editor has drafted this decision to help you prepare a revised submission.

Summary:

Stuart et al. present a compelling and comprehensive description of TE variation in *Arabidopsis thaliana*, highlighting the influence of TE polymorphism on expression of nearby genes and methylation of nearby regions. This paper persuasively argues that TE variation is the genetic basis for most DNA methylation polymorphisms and makes an important contribution to our understanding of the genomic impacts of TEs. However, the reviewers raised several points that should be addressed before publication. Most generally, this manuscript focuses a bit too much on the process and not enough on the biology. Dense statistical descriptions that will be difficult to understand for most readers are common, yet some results are explained either very broadly or not at all, and some important biological issues are ignored. A greater emphasis on biological meaning would substantially improve this paper.

Essential revisions:

1) The authors report that TEPID identified 300-500 insertions and 1000-1500 deletions per ecotype relative to Col. However, the overall number of insertions identified was 15,007 vs. 8,088 deletions. The reason that more deletions than insertions were identified per ecotype but fewer overall is that some of the deletions are present in nearly all or all tested ecotypes. Because this analysis is relative – insertions are both insertions in other ecotypes and deletions in Col, whereas deletions are deletions in other ecotypes and insertions in Col, such 'fixed deletions' are just insertion in Col. The authors get to this issue later in the analysis, but they don't explain the asymmetry. Why aren't there 'fixed insertions' relative to Col – meaning deletion in Col? Is TEPID unable to identify such events, or is there a biological meaning to this? An explanation would be very helpful.

2) The authors mention that TE insertions and deletions are both biased for pericentric regions, without discussing the observation that deletions are far more biased and what this likely means for actual TE insertion biases. The data strongly suggest that TE insertions are largely random, but are eliminated by selection from the chromosome arms. However, reading this paper, one could easily get the impression that TE insertion and deletion location biases are similar (which is stated in the Results), when they are actually not similar at all except in the most general sense.

3) C-DMRs and CG-DMRs are introduced without any description of the underlying biology (gene body vs. TE methylation; C-methylation mostly in TEs, CG mostly in genes) that would help to make sense of the different correlations of these DMRs with TE insertions. Gene body methylation doesn't even come up until the next-to-last Results section, and it's not explained there either. A keen reader might notice that CG-DMR distribution tracks genes, but to most the biological meaning of the presented analyses with respect to the two types of DMRs will be very obscure.

4) There is some concern about the inferences made about gene expression, as it seems that TE genes are not removed from the gene models used in the differential expression calls. Separating the impact of TEs inserting into TEs from the effect of TEs inserting into host genes would strengthen the argument that TEs are associated with extreme expression changes.

5) It may be useful to including some information about expectations of TE allele frequency in *Arabidopsis*, for example based on Helitron measurements by Hollister and Gaut (2007) (http://mbe.oxfordjournals.org/content/24/11/2515.long) and Gypsy measurements by Lockton and Gaut (2010) (http://bmcevolbiol.biomedcentral.com/articles/10.1186/1471-2148-10-10).

6) As any definition of a rare allele is somewhat arbitrary, the paper should be clearer about how TE variants are categorized into rare and common. Probably due to additions in the text, the definition of common (>3% MAF) comes in the subsection “Relationship between TE variants and single nucleotide polymorphisms”, long after Figure 2 is introduced. Can the common/rare definition either be added to the Figure 2 legend, or introduced when polarization is described? Similarly, it seems that the translation of MAF cutoffs to numbers of individuals are not consistent throughout the paper. While the definition of MAF cutoff states this is in >7 accessions, this doesn't obviously make sense with 217 accessions (shouldn't it be >=7?). Also, it seems that filtering reduces the number of accessions to 184, where 3% would be >5. Since there are so many low frequency TEs (Figure 2—figure supplement 2), small shifts in this definition might change conclusions qualitatively.

7) Gene At2G01360 used as an example in Figure 5 is highlighted in Figure 2 of Freeling et al. (2008) (http://genome.cshlp.org/content/18/12/1924.full) as transposing to this position in *Arabidopsis* by an unknown mechanism. Please comment about the possible relationship between this observation and the presence of a polymorphic TE within *Arabidopsis*.

[Editors’ note: a previous version of this study was rejected after peer review, but the authors submitted for reconsideration. The first decision letter after peer review is shown below.]

Thank you for submitting your work entitled "Population scale mapping of transposable element diversity reveals links to gene regulation and epigenomic variation" for consideration by *eLife*. Your article has been favorably evaluated by Detlef Weigel (Senior Editor) and four reviewers, one of whom is a member of our Board of Reviewing Editors. The following individual involved in review of your submission has agreed to reveal their identity: Magnus Nordborg (Reviewer #2).

Our decision has been reached after extensive consultation between the reviewers. Based on these discussions and the individual reviews below, we regret to inform you that your work will not be considered further for publication in *eLife* in its present form. We would, however, welcome a resubmission that substantively addresses reviewer concerns. Senior Editor Detlef Weigel suggests that you contact him directly if you have questions regarding the decision and recommendations. Magnus Nordborg is also happy to discuss his comments.

Although the reviewers found the subject of the manuscript interesting and clearly presented, three major deficiencies were identified:

1) The paper appears to conflate transposition with TE content variation, which are different things. TE variation – the actually measured feature – can occur through new insertions, but also through deletion of old copies. Given the large number of ecotype genomes and an understanding of the underlying phylogeny, new insertions could be distinguished from deletions of ancestral TEs in most cases. This would allow new insertion evens to be considered separately from deletions.

2) As explained by Magnus Nordborg in his review, the use of LD to measure TE insertion age is questionable and likely produces inaccurate measurements. This might explain why the conclusion reached in this manuscript that even recent TE insertions are biased for pericentric heterochromatin is inconsistent with studies of individual plant TEs that do not show such a bias. A more appropriate estimate of TE insertion age may be through measurement of allele frequency. Because estimated TE age is used in analyses throughout the manuscript, a robust measure is essential.

3) The section on seed and gamete methylation contains numerous questionable statements and analyses. Statements linking demethylation in pollen to imprinting are very confusing, because imprinting is associated with demethylation in the female gametophyte. The correlations with TEs demethylated (more accurately hypomethylated) in the CHH context in sperm ignore the global nature of this phenomenon, which affects most heterochromatic TEs. The small RNA analysis rests on the unwarranted assumption that the small RNAs are produced in the endosperm. The paper from which the small RNA data are derived makes a strong case that the small RNAs are derived from the maternal seed coat, which would provide a trivial explanation for the observation that TEs absent in the maternal genome have lower levels of small RNA in hybrid seeds. The most promising result – the correlation between TE content and methylation in Col x Cvi hybrids – didn't hold up in Col x L*er* hybrids.

We would welcome a resubmission that clearly identifies TE insertions, provides a robust measure of the false positive and false negative rates (please see Magnus Nordborg's review below), provides a robust measure of TE age, and ideally deepens the analysis of the correlations between TE variation, DNA methylation polymorphism, and gene expression as suggested by the reviewers.

Reviewer #2 (Magnus Nordborg):

This is a very clear and interesting paper. It is one of the first genome-wide descriptions of TE polymorphism (there will be more…), and, although the conclusions are a bit limited, it should be of broad interest. Most importantly, it really suggests strongly that methylation variation is mostly genetic.

I do have a few substantive concerns, however. The first concerns data quality, which has been a major reason these types of studies have not done before. Calling TE polymorphisms from short-read data is not trivial. What is done here is sensible, but I would still like to have some idea of error rates and biases. Using simulations (subsection “Computational identification of TE presence/absence variation”) is not QC – it is at best debugging of software. What is needed is comparison with real data, preferably a "gold standard". The obvious choice is the PacBio L*er* assembly, and it is used here, but it seems to me it is used in the wrong direction. The TE polymorphisms identified in the short-read data are confirmed in the PacBio data, rather than the other way around. It is the usual false-positive / false-negative issue that plagues all these data. What is done shows that the identified polymorphism are reliable, but it tells us nothing about what we don't see. Which leads to biases of all kinds. Why not do the right thing? And if there is a good reason for not doing the right thing, please justify this, and say a few words about how why the inevitable biases do not affect your results.

My second concern is the entire missing heritability section (Figure 2), which I think is based on two misunderstandings, one of linkage disequilibrium (LD), one of the nature of "missing heritability". Take LD first. It is theoretically impossible for TEs variants to show a different LD behavior than SNPs unless they are due to repeat mutations (which have to be very frequent indeed to affect LD). As long as identity in state = identity by descent, it doesn't matter how mutations arise. Of course it is possible that some of the TE indels could be non-unique, but there is no evidence for this in this paper (you could look for it: does the same insertion occur on unrelated haplotypes?). I'm fairly convinced that the difference is r^2^ distributions you see are largely due to differences in the allele frequency. To confirm this, compare r^2^ between TEs and flanking SNPs to r^2^ between a SNP and flanking SNPs after making sure that the polymorphism being tested has the same derived allele frequency. This can be done by binning. I think you will see that there is no difference. And if you do see a difference, it is probably due to some other artifact (genotyping error?), because, as I said, it is theoretically impossible for there to be a difference (unless repeat mutations are very different, selection is strong, etc.).

Don't say things like "this variant was classified as a young TE insertion, as it is not in linkage disequilibrium with surrounding SNP" – the younger a mutation is, the stronger the LD (because there has been no time for recombination). Sure, r^2^ will be low, but this is a trivial consequence of the definition of r^2^.

As I see it, the relevance of TEs in the "missing heritability" debate is simply that they could be an important source of rare deleterious alleles. Such alleles, regardless of whether they are TEs, SNPs, or CNVs, will have the property that they can be mapped in pedigrees in which they segregate, but will *individually* explain very little of the population variation and thus cannot be mapped using GWAS. This has nothing to do with LD: it is a simple consequence of their frequency. LD is a red herring in this context. I would get rid of Figure 2. and simply mention this in the Discussion.

But speaking of rare variants, you seem to argue against TEs having an effect, at least on transcript abundance, although you note that perhaps the rare variants are the ones that matter (subsection “TE variants affect gene expression”, first paragraph). Did you try testing this in aggregate (analogously to what human geneticists do to establish that rare alleles matter)? Rather than carrying out a SNP-by-SNP test, which clearly doesn't work when frequencies are less than 3% (is that 2 alleles?), ask whether these rare TEs are, in aggregate, more likely to show extreme phenotypes than two (or three, or whatever the right number is) randomly chosen individuals? You could calculate some kind of two-tailed rank statistic, and simply ask whether they are more likely to be extreme (feel free to contact me if this is not clear).

As I said from the outset, the strength of this paper is confirming that a large fraction of DMRs are due to TEs. This could be made even stronger:

A) When estimating the fraction based on published DMR data (subsection “TE variants drive DNA methylation differences between accessions”, end of first paragraph), why not distinguish between types of DMRs?

B) When looking for DMR around identified TEs (subsection “TE variants drive DNA methylation differences between accessions”, first paragraph), why not using random regions as control?

C) The correlations between TE and methylation are nice, but is there any clear example of strong methylation *without* a TE? This would be of obvious interest.

D) have you considered trans-acting TEs? Even if there is no TE variant in cis, there could be one in trans. How many more DMRs can you explain if you consider nearby TEs, variable or not?

Trans-acting TEs could also be important when interpreting the crosses, but I'm guessing other reviewers will comment more extensively on that.

Reviewer #3:

This is a very nice analysis of transposon activity in *Arabidopsis* accessions and will serve as a platform for similar analyses in more complex genomes.

Reviewer #4:

The manuscript by Stuart and colleagues describes the development of a software pipeline for detection of variations in transposons insertions (TEPID). The authors applied this pipeline to the available genomic sequences of over 200 *Arabidopsis* accessions. They verified efficiency and accuracy of the TEPID and estimated ages of transposons based on linkage disequilibrium between transposon and flanking sequences. In addition, they assessed the degree of transposon derived regulation of transcription of the neighbouring genes, which turned out to be surprisingly limited and only two experimentally verified examples could illustrate TE derived transcriptional suppression or activation. Interestingly, suppression of transcription of a gene encoding a LRR protein seems to cause a change in the level of resistance to a bacterial pathogen when accessions with or without the transposon insertion were compared. The authors also studied the influence of transposon insertions on the local variation in DNA methylation claiming that especially old insertions affect variation in DNA methylation. Finally, in reciprocal inter-accession crosses they examined the epigenetic interaction between accessions differing in the distribution of particular transposable elements, comparing loci with presence of corresponding transposons in one or both accessions.

This is an interesting manuscript.

---

## [Author Response]

Summary:

Stuart et al. present a compelling and comprehensive description of TE variation in Arabidopsis thaliana, highlighting the influence of TE polymorphism on expression of nearby genes and methylation of nearby regions. This paper persuasively argues that TE variation is the genetic basis for most DNA methylation polymorphisms and makes an important contribution to our understanding of the genomic impacts of TEs. However, the reviewers raised several points that should be addressed before publication. Most generally, this manuscript focuses a bit too much on the process and not enough on the biology. Dense statistical descriptions that will be difficult to understand for most readers are common, yet some results are explained either very broadly or not at all, and some important biological issues are ignored. A greater emphasis on biological meaning would substantially improve this paper.

We have made numerous changes throughout the text in order to place more emphasis on the biology rather than the process. Below we address each specific point raised by the reviewers.

Essential revisions:

1) The authors report that TEPID identified 300-500 insertions and 1000-1500 deletions per ecotype relative to Col. However, the overall number of insertions identified was 15,007 vs. 8,088 deletions. The reason that more deletions than insertions were identified per ecotype but fewer overall is that some of the deletions are present in nearly all or all tested ecotypes. Because this analysis is relative – insertions are both insertions in other ecotypes and deletions in Col, whereas deletions are deletions in other ecotypes and insertions in Col, such 'fixed deletions' are just insertion in Col. The authors get to this issue later in the analysis, but they don't explain the asymmetry. Why aren't there 'fixed insertions' relative to Col – meaning deletion in Col? Is TEPID unable to identify such events, or is there a biological meaning to this? An explanation would be very helpful.

We do in fact identify nearly fixed insertions that we classify as a Col-0 deletion. These are however a very small fraction of all the insertion variants identified, likely due to some bias in the TE variant discover method using the short read data. TE insertions relative to the reference genome appear to be slightly more difficult to identify that TE absences (as shown in our simulations, Figure 1—figure supplement 1), and any missed TE insertions will decrease the allele frequency for that variant, resulting in a reduced number of TE insertions that are at a high enough minor allele frequency to be subsequently classified as a TE deletion in Col-0. This effect is restricted to Col-0, as that is the only accession where all true TE deletions must be identified as TE insertions in other accessions in the population. As this bias is restricted to Col-0 (one accession out of the 216 examined), we do not believe this affects conclusions made in our manuscript. We have added the following sentence to our revised manuscript to address this question:

“High allele frequency TE presence variants relative to Col-0, representing true deletions in Col-0, were much more rare, with 97.8% of initial TEPID insertion calls being subsequently classified as true insertions. […] Accessions were found to contain on average ~240 true deletions and ~300 true insertions (Figure 2—figure supplement 5).”

We have added an additional plot showing the number of true deletions and true insertions detected for each accession as Figure 2—figure supplement 5:

To further clarify for readers, we have also added the following text:

“Although more TE absences were found on an accession-by-accession basis, overall TE presences were more common in the population as TE absences were often shared between multiple accessions, indicative of a TE insertion unique to the Col-0 reference genome.”

2) The authors mention that TE insertions and deletions are both biased for pericentric regions, without discussing the observation that deletions are far more biased and what this likely means for actual TE insertion biases. The data strongly suggest that TE insertions are largely random, but are eliminated by selection from the chromosome arms. However, reading this paper, one could easily get the impression that TE insertion and deletion location biases are similar (which is stated in the Results), when they are actually not similar at all except in the most general sense.

Our data do suggest that TE insertions are largely random, whereas TE deletions are more biased towards the pericentromeric regions, likely due to the distribution of old TEs throughout the genome. Thank you, we now realize that this point was not clear in our manuscript and have accordingly made clarifications to the text in order to make this important point more obvious to readers. Specifically, we have added the following text to the revised manuscript:

“While TE deletions were strongly biased towards the pericentromeric regions where TEs are found in high density, TE insertions had a more uniform distribution over the chromosome. This suggests that TE insertion positions are largely random but may be eliminated from chromosome arms through selection, and accumulate in the pericentromeric regions where low recombination rates prevent their removal (Figure 2).”

3) C-DMRs and CG-DMRs are introduced without any description of the underlying biology (gene body vs. TE methylation; C-methylation mostly in TEs, CG mostly in genes) that would help to make sense of the different correlations of these DMRs with TE insertions. Gene body methylation doesn't even come up until the next-to-last Results section, and it's not explained there either. A keen reader might notice that CG-DMR distribution tracks genes, but to most the biological meaning of the presented analyses with respect to the two types of DMRs will be very obscure.

We apologize for this oversight, and have added a brief explanation of the biological meaning of gene body methylation to the Introduction:

“In *Arabidopsis*, TEs are often methylated in all cytosine sequence contexts, in a pattern distinct from DNA methylation in other regions of the genome. Conversely, DNA methylation often occurs in gene bodies exclusively in the CG context and is correlated with gene expression, although this gene-body methylation appears dispensable (Bewick et al. 2016)”

We also draw attention to the CG-DMR distribution track in the third paragraph of the subsection “Abundant TE positional variation among natural *Arabidopsis* populations” to highlight the fact that the distribution of CG-DMRs closely follows the distribution of genes.

4) There is some concern about the inferences made about gene expression, as it seems that TE genes are not removed from the gene models used in the differential expression calls. Separating the impact of TEs inserting into TEs from the effect of TEs inserting into host genes would strengthen the argument that TEs are associated with extreme expression changes.

We have now repeated all analysis of the effect of TE variants upon gene expression removing all TE genes from the set of gene models used. The results are mostly the same, as most TE genes were lowly or not expressed in both cases of TE presence and TE absence, and so this change in the analysis does not affect any of our original conclusions.

5) It may be useful to including some information about expectations of TE allele frequency in Arabidopsis, for example based on Helitron measurements by Hollister and Gaut (2007) (http://mbe.oxfordjournals.org/content/24/11/2515.long) and Gypsy measurements by Lockton and Gaut (2010) (http://bmcevolbiol.biomedcentral.com/articles/10.1186/1471-2148-10-10).

We thank the reviewers for this suggestion, and have examined the allele frequencies for the *Basho* elements examined by Hollister and Gaut to compare with our data, as this dataset is the most straightforward to compare (the Lockton and Gaut paper uses TE display, which complicates the comparison of individual TE copy presence/absence calls). We found a weakly positive correlation between the TE occupation frequency for those elements examined by Hollister and Gaut, and the occupation frequency that we have estimated for the same elements using short read data in the larger population (Figure 2—figure supplement 9). We have added such information on the expected allele frequencies to the revised text:

“As certain TEs present in Col-0 have previously been genotyped in 47 different accessions, allele frequency data was available for some TEs (Hollister and Gaut 2007), and we compared these previous allele frequency estimates with our estimates based on the short read data. We found a weakly positive correlation (r^2^ = 0.3) between the previous allele frequency estimates for Basho family TEs and our allele frequency estimates, which may not be unexpected given the differing population sizes and TE variant detection methods used (Figure 2—figure supplement 9)”.

6) As any definition of a rare allele is somewhat arbitrary, the paper should be clearer about how TE variants are categorized into rare and common. Probably due to additions in the text, the definition of common (>3% MAF) comes in the subsection “Relationship between TE variants and single nucleotide polymorphisms”, long after Figure 2 is introduced. Can the common/rare definition either be added to the Figure 2 legend, or introduced when polarization is described? Similarly, it seems that the translation of MAF cutoffs to numbers of individuals are not consistent throughout the paper. While the definition of MAF cutoff states this is in >7 accessions, this doesn't obviously make sense with 217 accessions (shouldn't it be >=7?). Also, it seems that filtering reduces the number of accessions to 184, where 3% would be >5. Since there are so many low frequency TEs (Figure 2—figure supplement 2), small shifts in this definition might change conclusions qualitatively.

We apologize for this error, and the number of accessions used should read >6 (not >7). Furthermore, we now realize that the explanation of MAF cutoffs used may have been unclear in our original manuscript. As the LD analysis performed is very sensitive to having low SNP variation between accessions, some accessions were filtered out for that analysis, giving a set of 184 accessions. In this case, our 3% MAF cutoff amounted to >5 accessions. For all other analyses, we used the full set of accessions in order to maximize sample sizes, and so the cutoff for rare variants was >6 accessions. We have made alterations to the text in order to clarify this point, under the Methods section titled “TE variant / SNP comparison”. Furthermore, we have added a definition of the MAF used to the Figure 2 legend and again in the main text when the distinction between rare and common alleles is introduced (subsection “Abundant TE positional variation among natural *Arabidopsis* populations”, third paragraph).

7) Gene At2G01360 used as an example in Figure 5 is highlighted in Figure 2 of Freeling et al. (2008) (http://genome.cshlp.org/content/18/12/1924.full) as transposing to this position in Arabidopsis by an unknown mechanism. Please comment about the possible relationship between this observation and the presence of a polymorphic TE within Arabidopsis.

We thank the reviewers for pointing this out, and have added some discussion of the transposition events at this point to the text. Specifically:

“This TE insertion occurred at the border of a non-syntenic block of genes thought to be a result of a transposition event in *Arabidopsis* (Freeling et al. 2008). This transposition event likely predates the TE insertion discovered here, and it is interesting that multiple transposition events appear to have occurred in close proximity in the genome.”

[Editors’ note: the author responses to the first round of peer review follow.]

Although the reviewers found the subject of the manuscript interesting and clearly presented, three major deficiencies were identified:

1) The paper appears to conflate transposition with TE content variation, which are different things. TE variation – the actually measured feature – can occur through new insertions, but also through deletion of old copies. Given the large number of ecotype genomes and an understanding of the underlying phylogeny, new insertions could be distinguished from deletions of ancestral TEs in most cases. This would allow new insertion evens to be considered separately from deletions.

In our revised manuscript, we now distinguish between new TE insertions and deletions of ancestral TE copies. We used the minor allele frequency of TE variants to guide these classifications, and have carried out all subsequent analyses on these insertion and deletion groups separately, as they are fundamentally different events. We identify important differences between TE insertions and TE deletions in the patterns of DNA methylation surrounding these sites. While the level of DNA methylation surrounding TE insertions was often elevated, we do not see DNA methylation decrease following the deletion of a TE. Rather, these regions flanking the original insertion site remain highly methylated. We believe that making this distinction between TE insertions and TE deletions has greatly improved our manuscript, and thank the reviewers for their suggestion.

2) As explained by Magnus Nordborg in his review, the use of LD to measure TE insertion age is questionable and likely produces inaccurate measurements. This might explain why the conclusion reached in this manuscript that even recent TE insertions are biased for pericentric heterochromatin is inconsistent with studies of individual plant TEs that do not show such a bias. A more appropriate estimate of TE insertion age may be through measurement of allele frequency. Because estimated TE age is used in analyses throughout the manuscript, a robust measure is essential.

We agree that the use of LD as a measure of TE insertion age may produce inaccurate results, and have removed any statements linking TE LD with TE insertion age. Instead, we have focused on allele frequency as a proxy for TE insertion age, as suggested by the reviewers. We now analyse rare and common TE variants separately, and conclude that many of the TE variants we find are not tagged by SNPs.

3) The section on seed and gamete methylation contains numerous questionable statements and analyses. Statements linking demethylation in pollen to imprinting are very confusing, because imprinting is associated with demethylation in the female gametophyte. The correlations with TEs demethylated (more accurately hypomethylated) in the CHH context in sperm ignore the global nature of this phenomenon, which affects most heterochromatic TEs. The small RNA analysis rests on the unwarranted assumption that the small RNAs are produced in the endosperm. The paper from which the small RNA data are derived makes a strong case that the small RNAs are derived from the maternal seed coat, which would provide a trivial explanation for the observation that TEs absent in the maternal genome have lower levels of small RNA in hybrid seeds. The most promising result – the correlation between TE content and methylation in Col x Cvi hybrids – didn't hold up in Col x Ler hybrids.

We agree with the reviewers’ comments, and this section has been removed from the revised manuscript.

We would welcome a resubmission that clearly identifies TE insertions, provides a robust measure of the false positive and false negative rates (please see Magnus Nordborg's review below), provides a robust measure of TE age, and ideally deepens the analysis of the correlations between TE variation, DNA methylation polymorphism, and gene expression as suggested by the reviewers.

We believe each of these points have been addressed in the revised manuscript. We provide a robust measure of false positive and false negative rates, described in detail in the subsection “Computational identification of TE presence/absence variation” and subsection “TE variant simulation”, and find that our rate of false negatives is substantially lower than that of Quadrana et al., published recently in *eLife.* By using the same methodology as Quadrana et al. (2016), we enable a direct comparison between the two TE variant detection methods, and find that the method we have developed has a much greater sensitivity. This key difference allows us to detect 8x more TE variants, and this larger sample size further allows us to explore the effect of these TE variants at the level of individual accessions, rather than at the species level, as was the focus of Quadrana et al. (2016). To provide a measure of TE age, we have focussed on the minor allele frequency of TE variants, and make many comparisons between the rare and common alleles identified. We further deepened the analysis of correlations between TE variation, DNA methylation and gene expression by testing the effects of rare variants on both gene expression and DNA methylation, and we perform a genomewide scan for significant correlations between TE variants and genomewide patterns of DNA methylation, in order to identify any potential trans effects of TE variants on DNA methylation. Furthermore, we now analyse TE insertions and TE deletions separately, and find important differences between insertions and deletions in the effect such TE variants have on nearby patterns of DNA methylation.

Reviewer #2 (Magnus Nordborg):

This is a very clear and interesting paper. It is one of the first genome-wide descriptions of TE polymorphism (there will be more…), and, although the conclusions are a bit limited, it should be of broad interest. Most importantly, it really suggests strongly that methylation variation is mostly genetic.

I do have a few substantive concerns, however. The first concerns data quality, which has been a major reason these types of studies have not done before. Calling TE polymorphisms from short-read data is not trivial. What is done here is sensible, but I would still like to have some idea of error rates and biases. Using simulations (subsection “Computational identification of TE presence/absence variation”) is not QC – it is at best debugging of software. What is needed is comparison with real data, preferably a "gold standard". The obvious choice is the PacBio Ler assembly, and it is used here, but it seems to me it is used in the wrong direction. The TE polymorphisms identified in the short-read data are confirmed in the PacBio data, rather than the other way around. It is the usual false-positive / false-negative issue that plagues all these data. What is done shows that the identified polymorphism are reliable, but it tells us nothing about what we don't see. Which leads to biases of all kinds. Why not do the right thing? And if there is a good reason for not doing the right thing, please justify this, and say a few words about how why the inevitable biases do not affect your results.

We have now included a robust estimate of TE false negative calls, using the L*er* assembly and the same methodology as was recently published by Quadrana et al. To estimate the rate of false negatives for TE absence calls, we compared the set of TEs identified by Quadrana et al. as being absent from the L*er* genome (by BLAT alignment) to our set of TE absence calls for L*er* using TEPID. To estimate the rate of false negative TE insertion calls, we mapped Col0 short read data to the L*er* PacBio assembly, and then looked for read evidence at the sites containing a Col0specific TE insertion meeting the thresholds for a TE insertion call to be made by TEPID. For both TE absences and TE insertions, this resulted in a false negative estimate of approximately 10%. This is described in more detail in the last paragraph of the subsection “Computational identification of TE presence/absence variation” and subsection “TE variant discovery”, last paragraph.

My second concern is the entire missing heritability section (Figure 2), which I think is based on two misunderstandings, one of linkage disequilibrium (LD), one of the nature of "missing heritability". Take LD first. It is theoretically impossible for TEs variants to show a different LD behavior than SNPs unless they are due to repeat mutations (which have to be very frequent indeed to affect LD). As long as identity in state = identity by descent, it doesn't matter how mutations arise. Of course it is possible that some of the TE indels could be non-unique, but there is no evidence for this in this paper (you could look for it: does the same insertion occur on unrelated haplotypes?). I'm fairly convinced that the difference is r^2^ distributions you see are largely due to differences in the allele frequency. To confirm this, compare r^2^ between TEs and flanking SNPs to r^2^ between a SNP and flanking SNPs after making sure that the polymorphism being tested has the same derived allele frequency. This can be done by binning. I think you will see that there is no difference. And if you do see a difference, it is probably due to some other artifact (genotyping error?), because, as I said, it is theoretically impossible for there to be a difference (unless repeat mutations are very different, selection is strong, etc.).

We agree that the discussion of missing heritability in our manuscript was misguided, and have removed those statements. We now simply analyse TE LD in order to make the point that most of these variants are untagged by SNPs, and so some differences between accessions that we detect could not previously have been attributed to any genetic differences between accessions. One example of this is the TE insertion that we identify in *AtRLP18*, which is untagged by SNPs, and so the differing sensitivities of accessions to *Pseudomonas* infection would appear random in the absence of our TE variant data. We also include a comparison between LD state and minor allele frequency, and we see that the two are indeed well correlated, indicating that our TE variant data is likely free of bias.

Don't say things like "this variant was classified as a young TE insertion, as it is not in linkage disequilibrium with surrounding SNP" – the younger a mutation is, the stronger the LD (because there has been no time for recombination). Sure, r^2^ will be low, but this is a trivial consequence of the definition of r^2^.

We have now removed all statements linking TE LD state with TE variant age.

As I see it, the relevance of TEs in the "missing heritability" debate is simply that they could be an important source of rare deleterious alleles. Such alleles, regardless of whether they are TEs, SNPs, or CNVs, will have the property that they can be mapped in pedigrees in which they segregate, but will individually explain very little of the population variation and thus cannot be mapped using GWAS. This has nothing to do with LD: it is a simple consequence of their frequency. LD is a red herring in this context. I would get rid of Figure 2. and simply mention this in the Discussion.

We agree that the TE variants identified are simply a source of rare variants, and have removed any discussion of “missing heritability”.

But speaking of rare variants, you seem to argue against TEs having an effect, at least on transcript abundance, although you note that perhaps the rare variants are the ones that matter (subsection “TE variants affect gene expression”, first paragraph). Did you try testing this in aggregate (analogously to what human geneticists do to establish that rare alleles matter)? Rather than carrying out a SNP-by-SNP test, which clearly doesn't work when frequencies are less than 3% (is that 2 alleles?), ask whether these rare TEs are, in aggregate, more likely to show extreme phenotypes than two (or three, or whatever the right number is) randomly chosen individuals? You could calculate some kind of two-tailed rank statistic, and simply ask whether they are more likely to be extreme (feel free to contact me if this is not clear).

This is an excellent suggestion, and we have now carried out this analysis. We do find a clear enrichment for rare TE variants near genes with expression extremes in the population, suggesting that these variants do indeed have a strong impact on gene expression. Similarly, we find a strong enrichment for nearby rare TE insertions in accessions with high DMR DNA methylation. Rare TE deletions were enriched for both high and low DMR methylation ranks in the population, suggesting that the DNA methylation level at DMRs sometimes does not return to preinsertion levels following a TE deletion.

As I said from the outset, the strength of this paper is confirming that a large fraction of DMRs are due to TEs. This could be made even stronger:

A) When estimating the fraction based on published DMR data (subsection “TE variants drive DNA methylation differences between accessions”, end of first paragraph), why not distinguish between types of DMRs?

We now include both types of DMR in our analysis (CDMRs and CGDMRs). We find that the DNA methylation level at CDMRs is dependent on nearby TE presence/absence, but CGDMR methylation level is usually (but not always) independent from any nearby TE variants.

B) When looking for DMR around identified TEs (subsection “TE variants drive DNA methylation differences between accessions”, first paragraph), why not using random regions as control?

In this section, we aim to compare DNA methylation levels in the same region of the genome with the presence or absence of TE insertions, and so feel the comparisons made fit our purpose.

C) The correlations between TE and methylation are nice, but is there any clear example of strong methylation without a TE? This would be of obvious interest.

We do find that many CDMRs (and most CGDMRs) are not associated with a TE variant. We find that TEs can explain ~54% of the CDMRs. The remaining CDMRs that are apparently independent from TE presence/absence did not seem to have any defining features that we could discern (genomic location, distance to genes, distance to TEs). However, one of our key findings is that the level of DNA methylation surrounding TEs often remains high following a TE deletion, suggesting that DNA methylation, once established near a TE, often remains after the TE has been deleted. This suggests that more CDMRs could be a result of TE insertions than we find direct evidence for, if these TEs were deleted in all accessions analysed in this population. These CDMRs would then represent the epigenetic “scars” of past TE insertions. It is also possible that some DMRs are regulated in *trans*, and we performed a genomewide scan to identify these DMRs, discussed below.

D) have you considered trans-acting TEs? Even if there is no TE variant in cis, there could be one in trans.

This is an excellent suggestion. We have reanalysed our TE variant and DMR data to perform pairwise associations across the genome. We identified the previously noted strong local (cis) effect of TE variants upon DNA methylation, and also identified many putative transeffect associations in which a single TE variant displays a high level of association with hundreds of DMRs. We have substantially edited the manuscript to include a section describing our association results and highlight the possibility that specific TE variants may influence distant methylation states beyond local effects.

How many more DMRs can you explain if you consider nearby TEs, variable or not?

Of the CDMRs that were not within 1 kb of a TE variant, 3,701 (27% of all CDMRs) are within 1 kb of a fixed TE. Therefore, 81% of all CDMRs are within 1 kb of a TE when taking into account both variable and fixed TEs in the population. Of the remaining 19% of CDMRs, most are within genes or intergenic regions. This information has been added to the first paragraph of the subsection “TE variants explain many DNA methylation differences between accessions”.

Trans-acting TEs could also be important when interpreting the crosses, but I'm guessing other reviewers will comment more extensively on that.

As detailed above, the section has been removed from the revised manuscript.